# Landing force reveals new form of motion-induced sound camouflage in a wild predator

Kim Schalcher[1]*, Estelle Milliet[1], Robin Séchaud[1,2], Roman Bühler[3], Bettina Almasi[3], Simon Potier[4,5], Paolo Becciu[1†], Alexandre Roulin[1†], Emily LC Shepard[6†]

[1]Department of Ecology and Evolution, University of Lausanne, Lausanne, Switzerland; [2]Agroecology and Environment, Agroscope, Zurich, Switzerland; [3]Swiss Ornithological Institute, Sempach, Switzerland; [4]Department of Biology, Lund University, Lund, Sweden; [5]Les Ailes de l'Urga, Marcilly-la-Campagne, France; [6]Department of Biosciences, Swansea University, Swansea, United Kingdom

*For correspondence:
kim.schalcher@unil.ch

[†]These authors contributed equally to this work

Competing interest: The authors declare that no competing interests exist.

**Abstract** Predator-prey arms races have led to the evolution of finely tuned disguise strategies. While the theoretical benefits of predator camouflage are well established, no study has yet been able to quantify its consequences for hunting success in natural conditions. We used high-resolution movement data to quantify how barn owls (*Tyto alba*) conceal their approach when using a sit-and-wait strategy. We hypothesized that hunting barn owls would modulate their landing force, potentially reducing noise levels in the vicinity of prey. Analysing 87,957 landings by 163 individuals equipped with GPS tags and accelerometers, we show that barn owls reduce their landing force as they approach their prey, and that landing force predicts the success of the following hunting attempt. Landing force also varied with the substrate, being lowest on man-made poles in field boundaries. The physical environment, therefore, affects the capacity for sound camouflage, providing an unexpected link between predator-prey interactions and land use. Finally, hunting strike forces in barn owls were the highest recorded in any bird, relative to body mass, highlighting the range of selective pressures that act on landings and the capacity of these predators to modulate their landing force. Overall, our results provide the first measurements of landing force in a wild setting, revealing a new form of motion-induced sound camouflage and its link to hunting success.

## eLife assessment

This **fundamental** work substantially advances our understanding of animals' foraging behaviour by monitoring the movement and body posture of barn owls in high resolution and assessing their foraging success. With a large dataset, the evidence supporting the main conclusions is **compelling**. This work provides new corroboration for motion-induced sound camouflage and has broad implications for understanding predator-prey interactions.

## Introduction

Predation represents one of the strongest forms of selection in nature (*Grant and Clarke, 2001*; *Cook and Saccheri, 2013*; *Cuthill, 2019*; *Stevens and Merilaita, 2011*; *Hall et al., 2013*; *Dawkins and Krebs, 1979*). As a result, animals have evolved sophisticated adaptations to modify the sensory information they emit (*Stevens and Merilaita, 2011*; *Brooker and Wong, 2020*; *Ruxton, 2009*; *Garrouste et al., 2016*; *Caro et al., 2014*). Camouflage has been widely studied as an anti-predator

defence, with mechanisms including background matching, disruption, and self-shadow concealment facilitating predator avoidance (*Stuart-fox et al., 2006*; *Stevens et al., 2011*; *Ruxton et al., 2018*; *Ancillotto et al., 2022*). Predators also show adaptations to reduce detection by prey e.g., in their color, markings, and/ or behavior (*Pembury Smith and Ruxton, 2020*; *Théry and Casas, 2002*; *San-Jose et al., 2019*). However, in general, predator camouflage is far less understood due to the challenges of simulating predation in controlled settings (*Nebel et al., 2019*) and observing predation attempts in the wild (*Morisaka and Connor, 2007*). This has hindered our understanding of the evolutionary forces driving predator camouflage and explains why predator cues have yet to be linked to prey capture success.

Predation typically requires movements of a predator towards its prey, either during a pursuit or an ambush, which usually exposes chasing predators to detection (*Pembury Smith and Ruxton, 2020*; *Anderson and McOwan, 2003*). Indeed, motion makes individuals more conspicuous (*Hall et al., 2013*; *Stevens et al., 2011*; *Rushton et al., 2007*; *Regan and Beverley, 1984*). But motion also produces sound through the generation of vibrations and turbulence (*Larsson, 2012*; *Clark, 2016*), which can be detected by prey with acute hearing. Many predators alter their movements accordingly, for instance, by moving slowly during the pursuit, which may provide both acoustic and visual camouflage (*Ruxton, 2009*), particularly when combined with a background colour matching (*Pembury Smith and Ruxton, 2020*; *Anderson and McOwan, 2003*; *Zylinski et al., 2009*). While the direct link to hunting success remains unclear (*Pembury Smith and Ruxton, 2020*; *Mizutani et al., 2003*), selection should favour camouflage strategies that reduce sound emission in quiet environments. The resulting arms race may explain why many nocturnal species have acute senses of hearing, which they rely on to detect danger or prey (*Ruxton, 2009*; *Gerkema et al., 2013*; *Popper and Fay, 1997*).

The silent flight of owls is one of the most iconic examples of noise camouflage. Quiet flight is achieved through comb-like serrations on the leading edge of owls' wing feathers that break up the turbulent air and minimize associated sound production (*Ruxton, 2009*; *Clark et al., 2020*). This should provide advantages when hunting on the wing. However, most owls also launch attacks from perches, which involves moving from one perch to the next as they approach their prey (*Payne, 1971*; *Roulin, 2019*; *Taylor, 2004*). Landing also produces vibrations, and hence sound, with the intensity being proportional to the landing force (*Wernli et al., 2016*). In this dynamic sit-and-wait strategy, landing likely becomes a key element of the prey approach. We use high-frequency GPS and accelerometer data to investigate the landing dynamics of this sit-and-wait strategy in wild barn owls (*Tyto alba*). Specifically, we quantify whether the landing force varies with (i) the time until the hunting strike (i.e. hunting motivation), (ii) perch type (i.e. environmental context), and (iii) body mass, which varies between males and females (*Roulin et al., 2001*). Finally, we test the extent to which the magnitude of the landing force affects hunting success.

## Results

We used GPS loggers and accelerometers to record high-resolution movement data during two consecutive breeding seasons (May to August in 2019 and 2020) from 163 wild barn owls (79 males and 84 females) breeding in nest boxes across a 1000 km² intensive agricultural landscape in the western Swiss plateau. Of these individuals, 142 belonged to pairs for which data were recovered from both partners (71 pairs in total, 40 in 2019, 31 in 2020). The remaining 21 individuals belonged to pairs with data from one partner (11 females and 1 male in 2019; four females and five males in 2020).

### Measurement of landing force

We used the acceleration data to identify 84,855 landings. These were further categorized into perching events (n=56,874) and hunting strikes (n=27,981), depending on whether barn owls were landing on a perch or attempting to strike prey on the ground (*Figure 1A and B*, see methods for specific details on behavioural classification). We extracted the peak vectorial sum of the raw acceleration during each landing and converted this to ground reaction force (hereafter 'landing force' in Newtons) using measurements of individual body mass (see methods for detailed description).

Hunting strikes had landing forces over four times higher than perching events (*Figure 1C*, *Appendix 1—table 2*, *Appendix 1—table 3*; ratio: 4.5, z-ratio: 486.3, p<0.001). When converted to

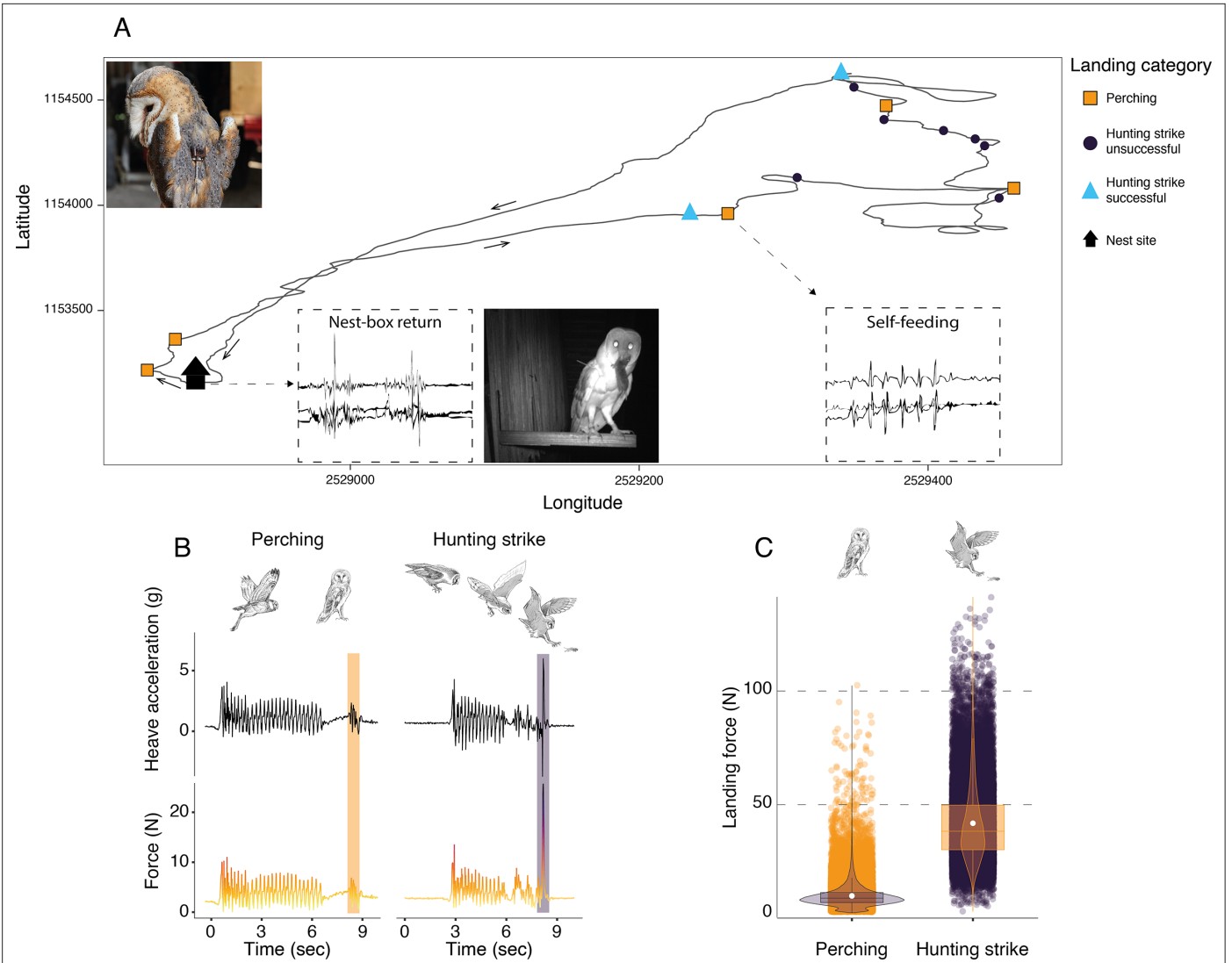

**Figure 1.** Sequence of perching and hunting strike events throughout a barn owl foraging trip. (**A**) GPS tracks (black line) during one complete foraging trip (duration = 73 min) performed by a female barn owl, with perching events (squares), unsuccessful (circles), and successful (triangles) hunting strikes. Black arrows indicate the flight direction. Successful hunting strikes were identified by the presence of self-feeding events (identified from the acceleration data), or by the direct return to the nest box (identified from the acceleration data and validated with the nest box camera footage). Inset panels show an example of the tri-axial acceleration signals corresponding to both nest-box return and self-feeding behaviours (see **Appendix 1— figure 3** for detailed representations). (**B**) The heave acceleration and the associated force during a perching event (highlighted in orange) and a hunting strike (highlighted in dark purple). (**C**) Variation in peak landing force for perching events (orange dots, n=56,874) and hunting strikes (dark dots, n=27,981). White dots show the estimated mean, and data distribution is represented by both violin and box plots. The owl picture at the top left of panel A is courtesy of J. Bierer, and owl drawings are courtesy of L. Willenegger, all used with permission.

multiples of body weight, hunting strikes had peak forces that were equivalent to approximately 13 times body weight, whereas perching events involved forces roughly three times body weight.

## Determinants of landing force

We conducted two sets of analyses to investigate factors that influence the variation in landing force in different landing contexts: perching events and hunting strikes.

Barn owls employing a sit-and-wait strategy land on multiple perches before initiating an attack, with successive landings reducing the distance to the target prey (*Figure 2C*). We analysed the landing forces involved in sequences of perching events in relation to perch type (poles, buildings, and trees: identified using GPS data) and the time before an attack (i.e. pre-hunt time: an indication

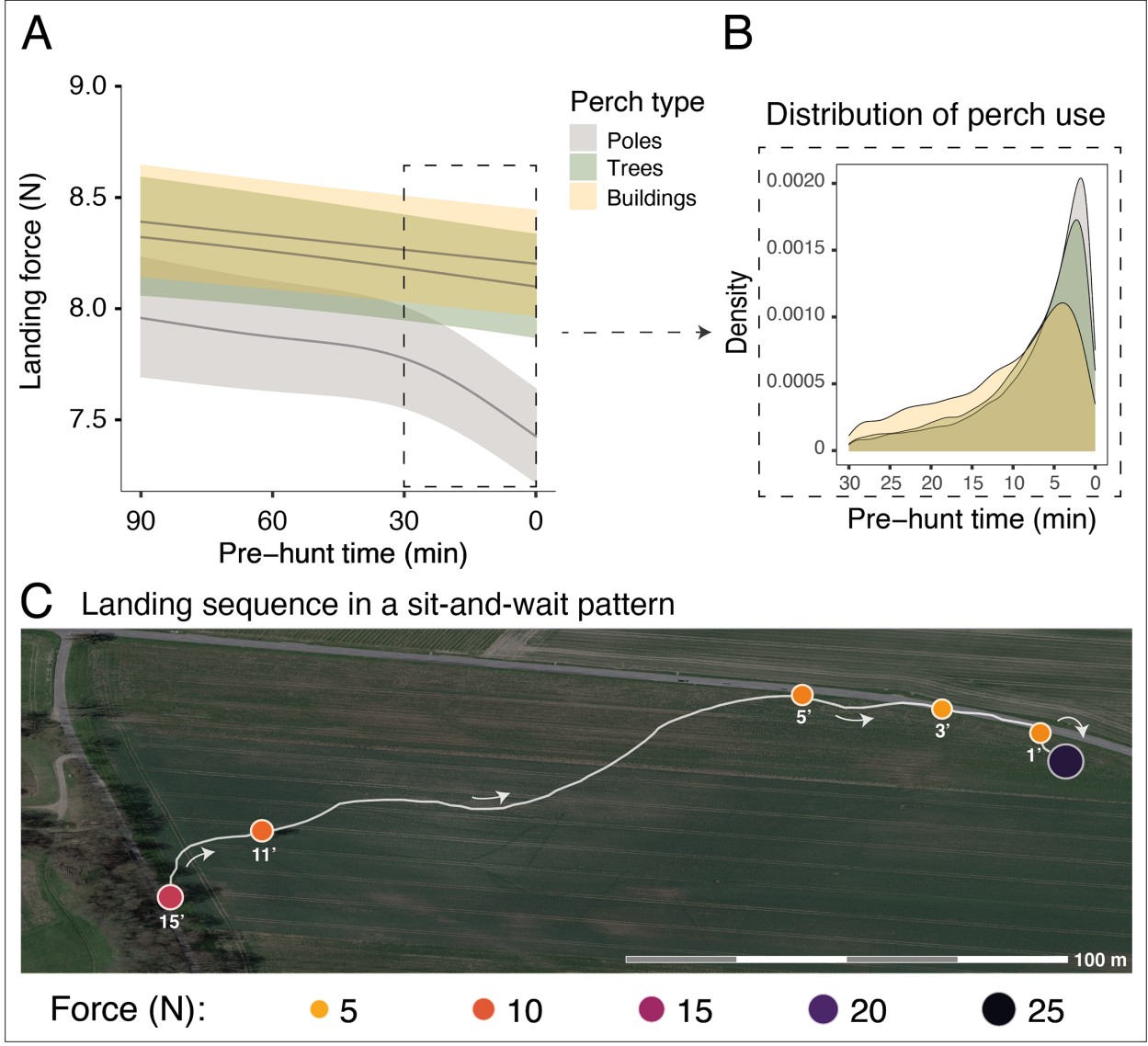

**Figure 2.** Sequential changes in perch type and landing force prior to hunting strikes during a sit-and-wait hunt. (**A**) Landing force during perching events (n=40,305) in relation to the time until the next hunting event and perch type. Each line represents the predicted mean for each perch type (here shown for males), with the 95% confidence intervals. (**B**) The selection of perch type in relation to time until the next hunting strike, highlighting the change in perch type that occurs ~10 min prior to a strike. (**C**) A sequence of perching events (orange circles) prior to a successful strike (purple circle) for a typical sit-and-wait hunt, showing the variation in peak landing force through time. White arrows indicate flight direction and numbers under each perching events indicate the time until the next hunting attempt (i.e pre-hunt time).

of hunting motivation). The most important predictor of landing force in perching events was perch type. Perching events on buildings were associated with the highest forces (8.96 N, CI: 8.90–9.01 N; *Appendix 1—table 4*, *Appendix 1—table 5*), closely followed by landing on trees (8.86 N, CI: 8.81–8.90 N; *Appendix 1—table 4*, *Appendix 1—table 5*). Poles were associated with the lowest landing force (8.33 N, CI: 8.28–8.38 N; *Appendix 1—table 4*, *Appendix 1—table 5*).

Importantly, within perch types, there was a reduction in landing force with time until the next hunting attempt, with the pattern differing with perch type (EDF$_{poles}$ = 4.22, p<0.001; EDF$_{buildings}$ = 1.00, p<0.001; EDF$_{trees}$ = 1.50, p=0.005; *Figure 2A*, *Appendix 1—table 4*, *Appendix 1—table 5*; n$_{tot}$ = 40,306 perching events; see *Appendix 1—figure 4* for the full representation and derivative plot). When barn owls perched on poles, the landing force showed a marked decrease in the last 30 min before the hunting strike, whereas landing force only showed a marginal linear reduction with time before the strike for landings on buildings (*Figure 2A*, *Appendix 1—figure 4*). Landing force did not

show any significant reduction with time for perching events on trees (*Figure 2A*, *Appendix 1—figure 4*, *Appendix 1—table 4*, *Appendix 1—table 5*). Our analysis also revealed a clear temporal pattern in the birds' use of perch types: owls launched more attacks from poles than from trees, with the fewest attacks launched from buildings (*Figure 2B*). The pattern of variation in landing force according to perch type and hunting motivation, and the pattern of perch use, were consistent for both males and females, despite females consistently exhibiting greater landing forces than males (*Appendix 1—table 4*, *Appendix 1—table 5*).

Additionally, our analysis of hunting strike force showed that both hunting strategy and success were related to strike force (*Appendix 1—table 6*, *Appendix 1—table 7*). When hunting on the wing, successful strikes involved greater forces than unsuccessful strikes ($n_{tot}$ = 24,464; successful strikes: $n_{succ}$ = 5830, 40.3 N, CI: 39.5–41.2 N; unsuccessful strikes: $n_{unsucc}$ = 18,634, 38.4 N, CI: 37.7–39.2 N). This was not the case when barn owls hunted from a perch ($n_{tot}$ = 3517; successful strikes: $n_{succ}$ = 1042, 38.8 N, CI: 37.7–40.0 N; unsuccessful strikes: $n_{unsucc}$ = 2475, 38.5 N, CI: 37.6–39.5 N).

## Sexual dimorphism and foraging behavior

Sexual dimorphism in body mass was marked among our sampled individuals. Males were lighter than females (84 females, average body mass: 322±22.6 g; 79 males, average body mass 281±16.5 g, *Appendix 1—figure 6*) and provided almost three times more prey per night than females (males: 8±5 prey per night; females: 3±3 prey per night; *Appendix 1—figure 7*). Males also displayed higher nightly hunting effort than females (Males: 46±16 hunting attempts per night, n=79; Females: 25±11 hunting attempts per night, n=84; *Figure 3A*, *Appendix 1—figure 8*). However, females were more likely to use a sit-and-wait strategy than males (females: 24%±15%, males: 13%±10%, *Appendix 1—figure 9*). As a result, the number of perching events per night was similar between males and females (Females: 76±23 perching events per night; Males: 69±20 perching events per night; *Appendix 1—figure 8*).

We conducted two different analyses to assess whether hunting strategies differed in success and efficiency (i.e. foraging trip duration). To assess the influence of strategy on foraging trip duration, we extracted the number of sit-and-wait hunting attempts and divided this by the total number of hunting attempts, and analysed this in relation to the trip duration (min) and sex. Our analysis showed that trip duration increased with the use of the sit-and-wait strategy (*Appendix 1—figure 11*, *Appendix 1—table 13*): barn owls that only used the sit-and-wait strategy (sit-and-wait frequency = 1) took an average of 15 min longer to provide prey to the nest than those that only hunted on the wing (sit-and-wait frequency = 0). Nonetheless, barn owls were more successful when using a sit-and-wait strategy, with success also varying with sex. Males were more successful than females, both for the sit-and-wait strategy (males: 34.5%, CI: 31.6–37.5%; females: 26.8%, CI: 24.4–29.2%, *Figure 3D*, *Appendix 1—table 8*, *Appendix 1—table 9*) and hunting on the wing (males: 26.1%, CI: 24.7–27.6%; females: 19.1%, CI: 17.7–20.5%, *Figure 3D*, *Appendix 1—table 8*, *Appendix 1—table 9*).

Landing force also varied with sex, with females generating landing forces that were 26% higher than males on average during perching events (*Figure 3C*, *Appendix 1—table 2*, *Appendix 1—table 3*) (females: 9.94 N, CI: 9.63 N–10.27 N; males: 7.91 N, CI: 7.65 N–8.18 N; ratio F/M: 1.26, 95% CI: 1.2–1.31). Males and females had similar landing forces during hunting strikes, with females generating forces that were only 6% higher than males on average (females: 40.8 N, CI: 39.49 N–42.18 N; males: 38.41 N, CI: 37.14 N–39.71 N; ratio F/M: 1.06, 95% CI: 1.01–1.11; *Appendix 1—figure 10*). However, when considered per unit of body mass (see methods), males exhibited lower forces than females when perching (males: 28.2 N/Kg, CI: 27.4 N/Kg–29.0 N/Kg; females: 30.9 N/Kg, CI: 30.1 N/Kg–31.8 N/Kg; *Appendix 1—figure 10*), but higher forces than females in hunting strikes (males: 136.2 N/kg, CI: 132.3 N/Kg–140.2 N/Kg; females: 126.4 N/kg, CI: 122.8 N/Kg–130.1 N/Kg, *Appendix 1—figure 10*).

Additionally. we performed two additional analyses to investigate potential variations in landing force and flight speed between sexes. We extracted the median ground speed (in m s$^{-1}$) of each flight prior to each hunting attempt and analysed this in relation to the sex of the individual. This showed that males flew slightly more slowly than females when searching for prey on the wing. Males flew slower than females by 0.23 ms-1 (Average flight speed males: 5.24 ms$^{-1}$, CI: 5.15 ms$^{-1}$–5.33 ms$^{-1}$; average flight speed females: 5.47 ms$^{-1}$, CI: 5.38 ms$^{-1}$–5.56 ms$^{-1}$, *Figure 3B*, *Appendix 1—table 12*).

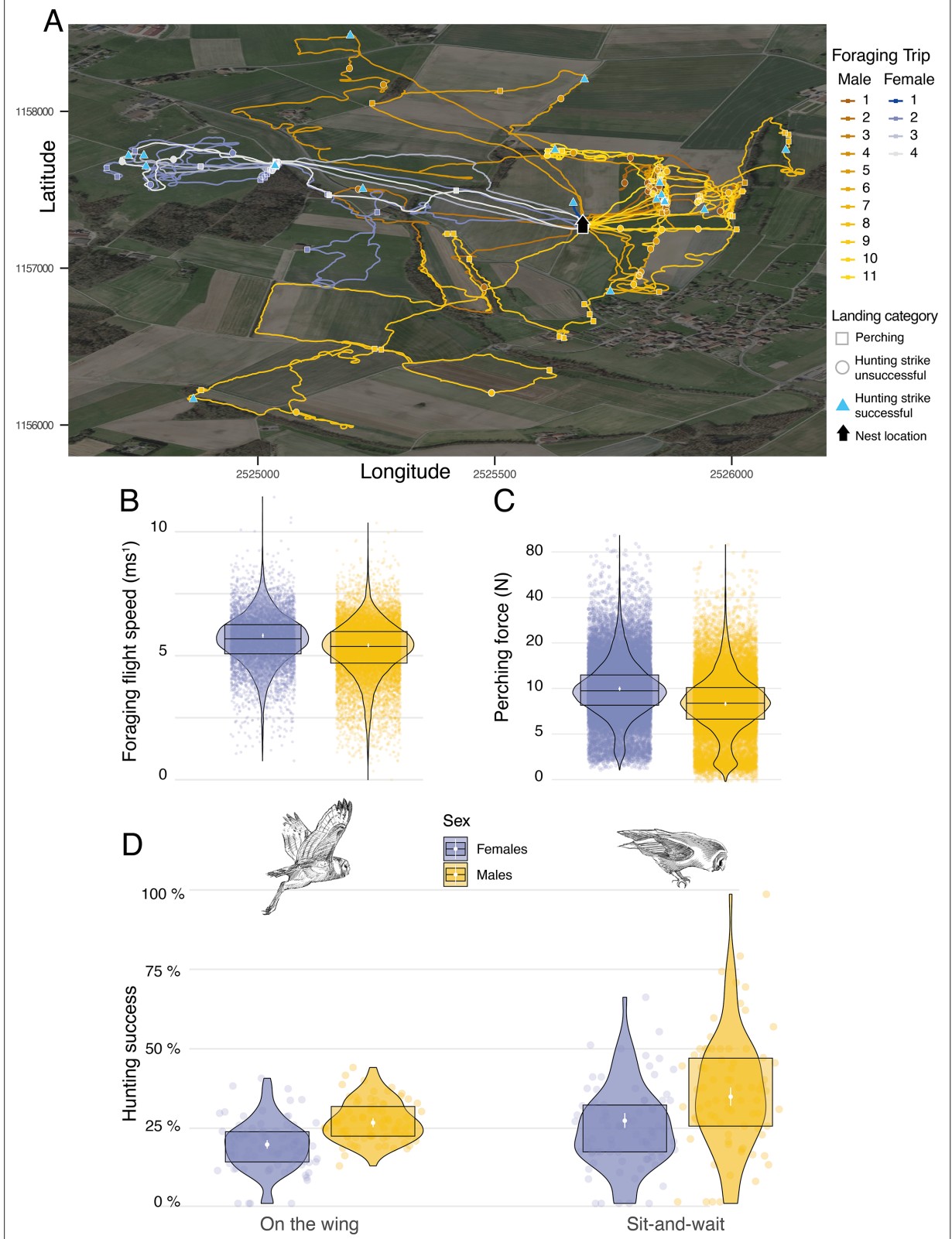

**Figure 3.** Sexual differences in foraging behaviour, landing force, and hunting success. (**A**) GPS tracks showing the foraging activities of a barn owl breeding pair during one complete night. Movement patterns of both male (yellow lines) and female (blue lines) are shown, with colour scale changing from the first trip of the night (foraging trip 1) to the last one (Male: $n_{max}$ = 11; Female: $n_{max}$ = 4). Perching events (squares), unsuccessful (circle) and successful (triangle) hunting attempts are shown for each foraging trip. (**B**) Variation in foraging flight speed for female (blues dots, n=9,223) and

*Figure 3 continued on next page*

*Figure 3 continued*

male (orange dots, n=18,019) barn owls (females: n=84; males: n=78). (**C**) Variation in peak landing force during perching events (perching force) for female (blue dots, n=30,378) and male (yellow dots, n=26,496) barn owls. (**D**) Variation in hunting success when barn owls hunted on the wing or used the sit-and-wait strategy for female (blue dots, $n_{on-the-wing}$=8,136,, $n_{sit-and-wait}$=1981) and male (yellow dots, $n_{on-the-wing}$=16,328,, $n_{sit-and-wait}$=1532) barn owls. For visualisation purposes, each dot shows the average hunting success of each individual for both hunting strategies. White dots and bars show the mean and the 95% CI around the mean, respectively, and data distribution is represented by both violin and boxplots. Owl drawings are courtesy of L. Willenegger, all used with permission.

## Pre-hunt landing force predicts hunting success for sit-and-wait strategy

Finally, we analysed whether the landing force in the last perching event before each hunting attempt (i.e. pre-hunt perching force) predicted variation in hunting success. Our results showed that hunting strategy was the strongest predictor of success (*Figure 4*, *Appendix 1—table 10*, *Appendix 1—table 11*, n=3040 hunting strikes from 151 individuals, see methods for details on data filtering). When hunting from the wing, the force applied during pre-hunt perching events had no effect on hunting success (*Figure 4*, *Appendix 1—table 10*, *Appendix 1—table 11*, odds ratio: 1.07, CI: 0.97–1.17, p=0.19). However, during sit-and-wait hunts, where the distance between the last perch and the prey is rather short (median distance 6.5 m, *Appendix 1—figure 5*), pre-hunt perching force predicted hunting success (*Figure 4*, *Appendix 1—table 10* , *Appendix 1—table 11*). When barn owls hunted directly from a perch, the chance of success decreased by 15% for every 1 N increase in pre-hunt perching force (odds ratio: 0.85, CI: 0.79–0.99, p=0.04). Perch type and wind speed were dropped from the final model after model selection (*Appendix 1—table 10*, *Appendix 1—table 11*).

## Discussion

Silent flight is considered crucial for owls hunting on the wing (*Clark et al., 2020*). But these predators also use a sit-and-wait strategy, with owls in this study achieving greater hunting success when launching attacks from a perch. Here, owls typically approach prey by moving between multiple perches. They must, therefore, avoid detection both in flight and as they land, as the benefits of silent flight may be negated if owls are detected during touchdown. We found that barn owls hunting from pasture poles reduced their landing force as they got closer to their prey. This suggests that soft landings are a novel form of acoustic camouflage, with predators reducing their motion-induced sound production in response to information on prey presence, which they gather as they move between perches in the final phase of the hunt. The landing force also affected the success of the subsequent strike, demonstrating the link between the predator camouflage and hunting success. However, the relatively low R-squared value (*Appendix 1—table 10*) suggests that hunting success is affected by additional factors such as prey behaviour, substrate type, and grass length (*Low-Décarie et al., 2014*).

Owls appeared to vary their perch use in relation to their motivation to hunt. For instance, barn owls that landed on a perch 30–90 min before a strike may have done so without the immediate intention of hunting. This phase was associated with greater use of buildings and trees, which are much higher structures than poles. Owls may, therefore, preferentially use these perches to rest or gather information over a wider area. The shift in predominant perch type to pasture poles 5–10 min before hunting strikes appears to represent a shift to periods of active prey searching. In some systems, the choice to hunt from pasture poles may be driven by prey availability, as pasture poles may be embedded in hedgerows or areas with longer grass where prey density might be higher. However, this is unlikely to be the case in our system as pasture poles generally occur in the middle of short-cropped grass (*Appendix 1—figure 2*). Instead, poles may offer advantages in being close to the ground, enhancing opportunities for owls to refine their estimates of prey location, prey type, or size. Indeed, the opportunity to gather information from perches could help explain the greater overall hunting success in attacks launched from a perch, compared to hunting on the wing.

In birds, landings are primarily governed by the need to maintain flight control and minimize the risk of injury (*KleinHeerenbrink et al., 2022*; *Lee et al., 1993*; *Paskins et al., 2007*),. For instance, Harris' hawks (*Parabuteo unicinctus*) landing in controlled conditions postponed the stall until they were as close to the landing perch as possible (*KleinHeerenbrink et al., 2022*). Such a strategy could serve two functions in barn owls using sit-and-wait hunting; minimizing both the energy dissipated

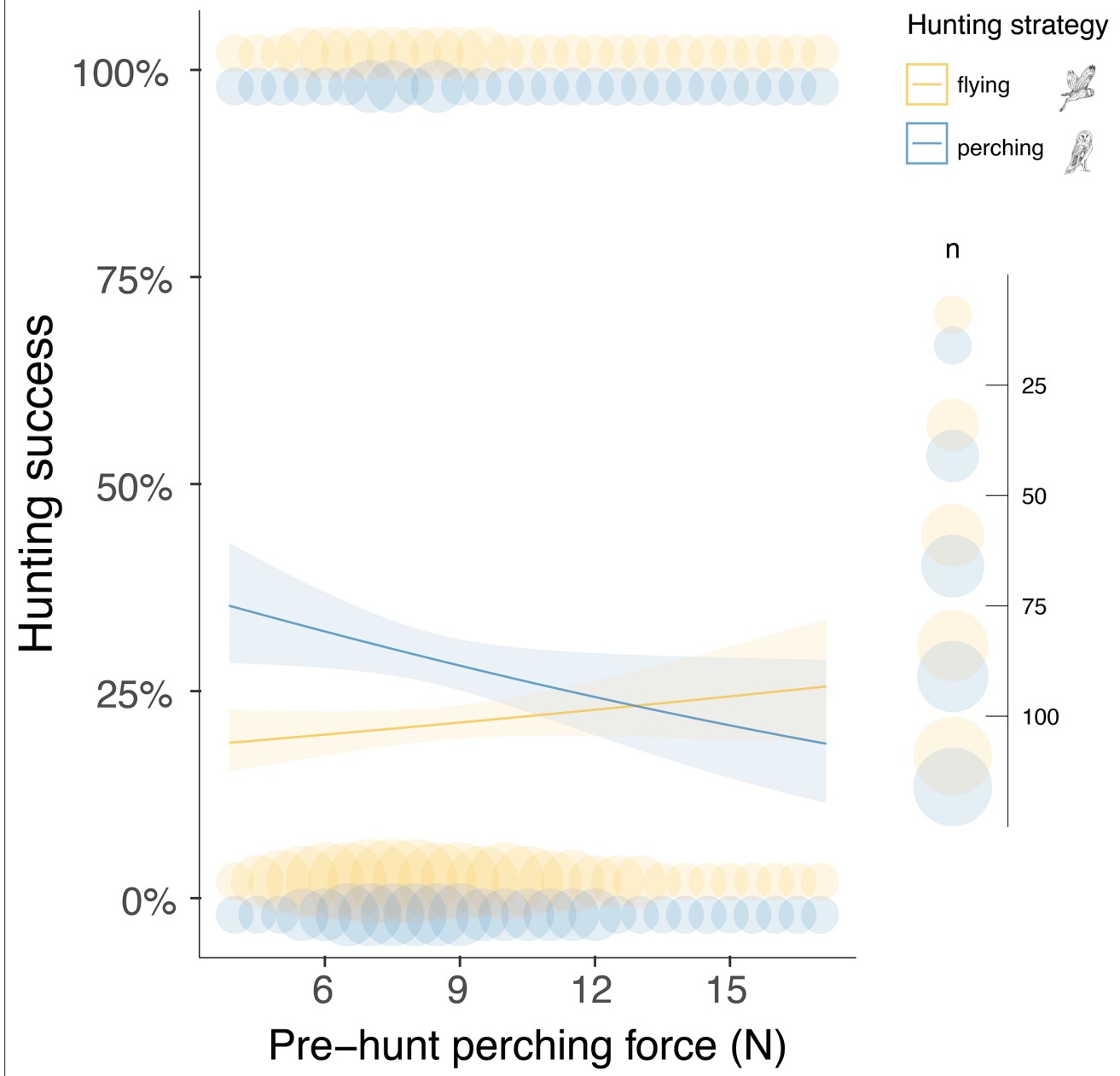

**Figure 4.** Pre-hunt perching force affects hunting success during sit-and-wait hunting. Variation in hunting success according to the pre-hunt perching force (N), depending on whether owls hunted on the wing (yellow) or using the sit-and-wait strategy (blue). Solid lines show the estimated means (averaged over both sexes), and the shaded area corresponds to the 95% confidence intervals around each mean. Blue and yellow circles show the force, recorded during the last perching event before hunting (pre-hunt perching, n=3040), pooled to the nearest integer N value for representation purposes, when hunting on the wing or using the sit-and-wait strategy, respectively. Circle size is related to the amount of data with the same value. The owl illustrations at the top right are courtesy of L. Willenegger, used with permission.

on impact and the associated sound production (**Wernli et al., 2016**). This raises the question of why owls would ever land with anything above the minimal force. To date, almost all studies have examined landings in controlled conditions (**KleinHeerenbrink et al., 2022**; **Roderick et al., 2017**), yet in the wild, birds are faced with a range of perch types and landing conditions. Perch characteristics are likely to play a pivotal role, as forces tend to be absorbed to a greater degree by compliant substrates (**Demes et al., 1995**). In support of this, landings on buildings were associated with the highest mean forces, and higher forces than tree branches (**Appendix 1—table 4**), which would be more compliant, with the extent varying with branch type and diameter (**Paskins et al., 2007**). It was, therefore,

notable that forces were lowest for landings on poles, which, like buildings, are rigid. However, poles occur in open habitats (*Appendix 1—figure 2*), providing a predictable landing surface that can be approached from all directions, facilitating control through optimal use of the wind vector. Landing force may, therefore, be influenced by the access options as well as the substrate type. There may also be greater incentive to reduce landing force on poles, since they are close to the ground and sound attenuates with distance (*Clark et al., 2020*; *Wahlberg and Larsen, 2017*; *Larsen et al., 2017*).

The biggest difference in landing force was observed between perching events and hunting strikes. Strike forces in our study are the highest recorded in any bird, relative to body mass, with maximal force reaching more than 34 times the body weight (100 N). This exceeds estimates previously reported for captive barn owls (*Usherwood et al., 2014*), and the kicking strike of the secretary bird (*Sagittarius serpentarius*) that reached an average of 5.1 times body weight (*Portugal et al., 2016*). Unlike secretary birds, whose kicking strength depends solely on the muscular power of their lower limbs, owls use the dynamics of their entire body in flight. While this likely minimizes the chances of prey escape, it is also associated with a potential risk of injury (*KleinHeerenbrink et al., 2022*; *Provini et al., 2014*). Our results likely underestimate the true peak forces, as acceleration was recorded at 50 Hz (for reference, data on force development in controlled car crashes are typically recorded at >2 kHz). Nonetheless, our data can still provide new insight into the selective pressures that have influenced owl morphology. Indeed, the lower limbs of owls allow for the dual function of absorbing shock during pre-hunt perching and generating extremely powerful hunting strikes.

We find that males and females had very similar strike forces, despite their substantial difference in body mass. This indicates that there might be a selective pressure for a minimum strike force, which males may generate by increasing or maintaining their flight speed prior to a strike to a greater extent than females. Males had a lower flight speed during prey searching, most likely due to their lower body mass (*Pennycuick, 2008*). While the difference in flight speed was relatively small, slower flight could still have advantages in (i) providing additional time to localize prey, and (ii) enabling birds to manoeuvre into the strike phase (*Amélineau et al., 2014*; *Machovsky-Capuska et al., 2012*). This may help explain why males have higher success when hunting on the wing. Males also showed greater hunting success than females in the sit-and-wait strategy. Here, a lower body mass could also provide advantages by facilitating lower impact, and hence quieter landings.

Given that sit-and-wait hunting is associated with higher success, why do male barn owls not use this strategy more (it was associated with less than 10% of hunting attempts)? Male barn owls engage in intense hunting activity in the breeding season, providing over 15 prey per night in our study. Our results showed that foraging duration increases with the use of the sit-and-wait strategy. Thus, the time required to capture prey appears to be the key element influencing the choice of hunting strategy in males. Females provide fewer prey items and the additional time required for sit-and-wait hunting may, therefore, be less of a constraint. Furthermore, sit-and-wait hunting may require less flight time and hence effort, which is likely to be particularly advantageous for females due to their greater body mass (and higher flight costs per unit mass).

In conclusion, we use high-frequency movement data to propose a novel form of acoustic camouflage and demonstrate that the magnitude of predator cues can influence hunting success (*English et al., 2024*). Minimizing landing force, and associated sound production is likely to be particularly pertinent for nocturnal predators, which operate in quiet environments and target prey with an acute sense of hearing (*Gerkema et al., 2013*; *Popper and Fay, 1997*; *Webster and Plassmann, 1992*). Importantly, the ability to minimize landing force was modulated by the perch characteristics, providing a potential link between landing impact and habitat characteristics. This suggests there could be spatial patterns in the effectiveness of acoustic camouflage and, ultimately, hunting success. The availability of different perch types could, therefore, be an additional, and previously unrecognized, aspect of habitat and territory quality, and, in this case, one that is strongly linked to land-use practices.

## Materials and methods
### Study area and tag deployment
Data were collected from wild barn owls breeding in nest boxes across the Western Swiss plateau, an area of 1000 km$^2$ characterized by an open and largely intensive agricultural landscape (*Almasi*

et al., 2015). Over 380 nest boxes were checked for barn owl clutches between March and August in 2019 and 2020, following Frey and colleagues' protocol (*Frey et al., 2011*). During the two breeding seasons, 163 breeding barn owls (84 females; 79 males) were equipped with data loggers (2019: 43 males and 49 females; 2020: 36 males and 35 females, *Appendix 1—figure 1*).

Adult barn owls were captured at their nest sites approximatiely 25 days after the first egg hatched using automatic sliding traps that are activated when birds enter the nest box. AXY-Trek Mini loggers (Technosmart, Rome, Italy) were attached as backpacks (*Figure 1A*) using a Spectra ribbon harness (Bally Ribbon Mills, USA). These units include a GPS, set to record animal location at 1 Hz, 30 min before sunset until 30 min after sunrise, to get the full nightly activity period. The loggers also include a tri-axial accelerometer, which records acceleration continuously at 50 Hz (recording range ±16 g, 10-bit resolution). After 10 days (±2 days), loggers were recovered by recapturing adult barn owls at their nest sites, again using automatic sliding traps, with data recorded for 5 nights on average (±1 night). Owls were weighed at both visits and the averaged body mass from the two measurements was used for later analysis. Each device weighed on average 12.4±0.1 g, which corresponds on average to 4% of the barn owl's total body mass (min = 3%, max = 5%, female average body mass: 322±22.6 g; males average body mass 281±16.5 g) and, therefore, never exceeded the limit of 5% of the bird's body mass (*Fair and Jones, 2010*).

In parallel to each logger deployment, motion-sensitive camera traps (Reconyx HC500 hyperfire, resolution of 3.1 megapixel) were positioned at the entrance of all nest boxes to document when animals returned to the nest with prey (*Figure 1A*). Camera traps were scheduled to record bursts of three pictures when motion was detected. Moreover, wind data were collected using portable weather stations (Vantage Vue, Davis Instruments Corp.) mounted 2.0 m from the ground (standard anemometer measurement height) within 100 m of each nest. Wind speed and direction were recorded every 10 min.

## Behavioural classification

We used Boolean-based algorithms (*Wilson et al., 2018*) to classify flight, landing, hunting strikes, and self-feeding from the onboard acceleration and GPS data (see below). Behaviours were summarized in 1 s intervals and linked to the closest GPS location in time. Flight, hunting, and self-feeding behaviours were ground-truthed using video footage of two captive barn owls equipped with the same data loggers. Further validations were undertaken for hunting behaviour (detailed below).

Behavioural classifications used the raw acceleration data, the vectorial dynamic body acceleration (VeDBA) (a summary metric of body motion), and body pitch angle (*Appendix 1—figure 3*, *Appendix 1—table 1*). VeDBA was derived by smoothing the raw acceleration data over 0.5 s (the period of two complete wingbeat cycles), to estimate the static/gravitational component, subtracting this from the raw acceleration in each of the three acceleration channels (*Shepard et al., 2008*), and calculating the vectorial sum from the resulting 'dynamic' components. Pitch angle was derived using the arcsine of the static acceleration in the heave axis (*Wilson et al., 2008*; *Shepard et al., 2010*) and smoothed over 1 s.

Flights were identifiable from the acceleration data as periods of take-off, travelling, and landing (*Appendix 1—figure 3A*). Take-offs were characterized by a switch from a standing to a horizontal posture (Δ pitch angle >–10 °) and high-amplitude VeDBA (>1 g) (*Usherwood et al., 2014*). Travelling flight was associated with smoothed VeDBA values >0.1 g, and body pitch values <30 °. Finally, landings were identifiable as changes from low to high pitch angles (Δ pitch angle >10 °) and a typical final spike in all three acceleration axes (VeDBA >1 g). Periods that did not correspond to flight were categorized as stationary behaviour.

Landings were further classified as either perching events, where owls landed on a perch prior to a hunting attempt, or hunting strikes/prey capture attempts (*Figure 1*). Landing types were categorized using the rate of change in pitch angle (strikes: Δ pitch angle >6 °) and the amplitude of the peak acceleration (strikes: Δ VeDBA >1.3 g) generated by the impact with the prey/ground, which were both much greater for hunting strikes than perching events (*Appendix 1—figure 3B*). Hunting strikes were classified using the Boolean-based classification algorithm (*Appendix 1—table 1*), whereas perching events were identified as the termination of flights that did not end with a hunting strike.

Owls hunt to provision themselves and their offspring. Self-feeding was evident from multiple and regular acceleration peaks in the surge and heave axes (resulting in peaks in VeDBA values >0.2 g

and <0.9 g, *Appendix 1—figure 3D*), with each peak corresponding to the movement of the head as the prey was swallowed whole. Prey provisioning events were identified from variations in the sway, corresponding to the owl walking inside the nest box (*Appendix 1—figure 3C*). Both start and end phases of the nest box visits were characterized by a rapid change in the pitch angle (enter: Δ pitch angle <−1.5 °; exit:<0.5 °) along with an increase in the heave and VeDBA values (enter: Δ VeDBA >0.5 g; exit: Δ VeDBA <−0.9), as owls leapt in/out of the nest box. Successful provisioning hunts were further confirmed using nest box camera data when available and, in all cases, by manually checking that the GPS data matched the nest site to identify cases where the owls returned with a single prey for their offspring. Unsuccessful strikes were, therefore, inferred from identified hunting strikes that were not followed by a provisioning to the nest and/or self-feeding event (*Figure 1A*).

## Data processing

Data from the onboard accelerometers can be used to estimate landing force during perching and hunting strikes (*Figure 1B*), as force is equal to the product of mass and acceleration. To estimate landing force, we extracted the peak vertical component of the ground reaction force in Newtons (N) for every landing event, taking the maximum value of the vectorial sum of the raw acceleration (in units of gravitational acceleration, *g*), multiplying this by the body mass of the bird (in kg) (*Pouliot-Laforte et al., 2014*; *Banerjee et al., 2014*).

Hunting strikes were categorized according to whether owls hunted on the wing or from a perch to assess factors affecting the landing force of perching events involved in the sit-and-wait strategy. We, therefore, considered that owls were using the sit-and-wait strategy if they flew for a maximum of 1 s before the strike (corresponding to c.a. 6.5 m from the last perch). Hunting on the wing was defined as cases when birds flew for at least 5 s prior to the strike (c.a. 81.7 m from the last perch). Hunting strikes that did not fit into either category (8% of all hunting strikes) were excluded from the dataset. When barn owls were hunting on the wing, we also estimated foraging flight speed by extracting the median ground speed (in ms$^{-1}$) over the last 20 s preceding each hunting strikes.

Finally, perch type was estimated by extracting the median location of each perching event. The habitat within 2 m was then classified according to the main perch type available: trees, roadsides, and pasture poles (hereafter referenced as 'poles'), and buildings, and assigned as the perch type for each perching event. Habitat categories (roads, settlements, single trees, forest) were provided by the Swiss TLM3d catalogue (Swiss Topographic Landscape Model, resolution 1–3 m depending on the habitat feature) and habitat data were provided by the 'Direction générale de l'agriculture, de la viticulture et des affaires vétérinaires (DGAV)' and the 'Direction des institutions, de l'agriculture et des forêts (DIAF),' for states of Vaud and Fribourg, respectively.

## Statistical analyses

We first assessed how landing force varied between hunting strikes and perching events, before evaluating the factors that explained variation within each category. This excluded perching events made when owls were loaded with prey, where the landing force will likely be influenced by the extra mass carried.

We fitted a linear mixed model (LMM) of the landing force (log-transformed) where fixed factors included the landing context (a two-level factor: hunting strike or perching event), the sex of the individual (a two-level factor: Female and Male) and their interaction. Sex was included in the model to control for sexual differences in foraging strategy as well as a sexual dimorphism in body mass (*Roulin, 2019*). The model included bird ID as a random intercept to account for repeated measurements of the same individual over multiple nights, and night ID (nested in bird ID) to account for repeated measurement of the same individual within the same night. The same random effect structure was applied to all the following LMs and GLMs as they were fitted to dataset of similar grouping structure. We also fitted a LMM of the landing force during hunting strikes (log-transformed). Fixed factors in the model included hunting success (a two-level factor: successful and unsuccessful), the hunting strategy (a two-level factor: perching and flying), and their interaction. Sex was also included as a fixed factor.

We next fitted a generalized additive mixed-effects model (GAMM) to assess how the landing force (log-transformed) varied between perching events. Specifically, we examined whether this was affected by the physical environment (perch type, wind), or motivation (owls can perch for long periods between hunts, and the most pertinent currency determining landing force may, therefore,

vary between periods of resting and active searching). Time until the next hunting strike was extracted for every perching event and included as a continuous fixed covariable in the model. An interaction between a smoothed function of the time until the next hunting strike and perch type was also included, using a thin plate regression spline and the 'by' condition, with the number of bases per smooth term (k) set at a conservative value of 9. The sex of the individual, windspeed, and perch type (a three-level factor: pole, tree, building) were included as linear predictors in the model. The model included the random intercept effect of bird ID (included with bs='re' in a smooth function).

Our GAMM of landing force showed that owls perched more softly the closer they came to the next hunting strike. To identify periods when there was a significant change in landing force, we calculated the first derivative $f'(x)$ of the estimated smoothed relationship between the time to the next strike and the peak landing force, according to each perch type, to highlight significant periods of positive or negative relationships (*Simpson, 2018*; *Becciu et al., 2023*). Periods of significant change were identified as those time points where the simultaneous confidence interval on the first derivative does not include zero.

Finally, we performed a set of analyses to investigate how hunting success varied with sex and hunting strategy, and most specifically whether success might be influenced by the landing force involved during perching events. To study how hunting success overall varied with sex and hunting strategy, we ran a first generalized linear mixed-effect model (GLMM) with hunting success as binary response variable (1=successful, 0=unsuccessful). In this first model, the sex of each individual, the hunting strategy, and their interaction were included as fixed effects.

Then, we fitted a second GLMM with hunting success as a binary response variable to specifically investigate whether the landing force applied in the last perching events would influence the success of the following hunting attempts. Hypothesizing that landing force might affects barn owl detectability, we only selected hunting strikes that were immediately preceded by a perching event (hereafter pre-hunt perching). We also selected hunting strikes that occurred <90 s after the last perching event to maximize the probability of capturing a response to the pre-hunt perching force. The threshold of 90 s corresponded to the lower tercile of the distribution of time differences between perching and hunting strikes. The fixed effects included in this second model were pre-hunt perching force (i.e the force applied during perching events directly preceding each hunting attempt), hunting strategy and their interaction. The sex of the individual, windspeed, and the interaction between sex and hunting strategy were also included as fixed effects in the model.

In birds, body mass usually influences flight speed (*Pennycuick, 2008*). We, therefore, hypothesized that the sexual dimorphism in body mass present in barn owls' population might influence the speed at which males and females would fly when foraging on the wing. This could in turn impact on their ability to locate and target prey on the ground and, therefore, ultimately influence hunting success when hunting on the wing. To test this hypothesis, we fitted a second LMM with foraging flight speed as continuous response variable and the sex of each individual as fixed effect.

Finally, we fitted a LMM to assess how preferences of a given hunting strategy might affect barn owls foraging trip duration. The model included the foraging trip duration (min) as response variable. The model also included the frequency of use of the sit-and-wait strategy (number of hunting attempts in the sit-and-wait per trip divided by the total number of hunting attempts per trip), the total number of hunting attempts per trip, and the sex as predictors.

All statistical analyses were conducted with R 4.0.5 (R Core Team, Vienna, Austria), with RStudio (RStudio Team, 2020) as graphic user interface. LMMs and GLMMs were fitted with the functions *lmer* and *glmer*, respectively, implemented in the package 'lme4' (R package v1.1–27.1) (*Bates et al., 2015*) and we used the package 'lmerTest' (R package v3.1–3) (*Kuznetsova et al., 2017*) to estimate p-values. GAMM model was fit using the *gam* function from the package 'mgvc' (R package v1.8–34) (*Wood, 2011*; *Wood, 2017*; *Wood, 2004*). For all models, linear predictors were centered and scaled to mean zero and units of standard deviation (i.e. z-scores) to ensure comparability among variables. We selected the optimal structure of the fixed component of each models using a multi-model selection framework ranking the selected models according to the Akaike information criterion (*Burnham and Anderson, 2002*; *Burnham et al., 2011*), using an automated stepwise model selection procedure in which models are fitted through repeated evaluation of modified calls extracted from the model containing all the meaningful variables, corrected for small sample sizes (AICc) (*Sugiura, 1978*). The final models were chosen as the best models among the candidate models within ΔAICc <2, that

was always relatively low (between 1 and 4) (see *Appendix 1—table 3, 5, 7, 9 and 11*). Additionally, we performed pairwise comparisons using the *emmeans* function from the package 'emmeans' (R package v1.6.0) (*Lenth et al., 2021*) to further assess differences between predictors level. Models were fitted, checked for collinearity between predictors, and assumptions were verified by visually inspecting residual diagnostic plots. Descriptive statistics are reported as Mean ± SD, unless specified otherwise.

## Acknowledgements

This study was supported by the Swiss National Science Foundation (grants no. 31003A_173178). We thank A P Machado; L Ançay, N Külling, A-C Heinz, M Froehly, M Calvani, N Sironi, L Legrand, D Zurkinden, R Allemand and L Hulaas for their help in collecting field data; P Potier and "les Aigles de l'Urga" for their help in behavior calibration with captive barn owls; R P Wilson and W Allen for their expertise and assistance throughout all aspects of our study and for their help in writing the manuscript; L Willenegger for providing barn owls drawings and J Bierer for barn owl picture.

## Additional information

### Funding

| Funder | Grant reference number | Author |
| --- | --- | --- |
| Swiss National Science Foundation | 31003A_173178 | Alexandre Roulin |

The funders had no role in study design, data collection and interpretation, or the decision to submit the work for publication.

### Author contributions

Kim Schalcher, Conceptualization, Data curation, Software, Formal analysis, Investigation, Visualization, Methodology, Writing – original draft, Writing – review and editing; Estelle Milliet, Formal analysis, Visualization, Writing – original draft, Writing – review and editing; Robin Séchaud, Conceptualization, Writing – original draft; Roman Bühler, Investigation, Visualization; Bettina Almasi, Resources, Funding acquisition, Writing – original draft; Simon Potier, Conceptualization, Methodology, Writing – original draft; Paolo Becciu, Supervision, Visualization, Methodology, Writing – original draft, Writing – review and editing; Alexandre Roulin, Conceptualization, Resources, Supervision, Funding acquisition, Writing – original draft, Project administration; Emily LC Shepard, Conceptualization, Software, Supervision, Validation, Investigation, Visualization, Methodology, Writing – original draft, Writing – review and editing

### Author ORCIDs

Kim Schalcher ⓘ https://orcid.org/0000-0003-0719-2765
Roman Bühler ⓘ https://orcid.org/0000-0002-4883-4923
Paolo Becciu ⓘ http://orcid.org/0000-0003-2145-6667
Alexandre Roulin ⓘ https://orcid.org/0000-0003-1940-6927
Emily LC Shepard ⓘ https://orcid.org/0000-0001-7325-6398

### Ethics

This study meets the legal requirements of capturing, handling, and attaching animal tracking devices to barn owls in Switzerland from the Department of the Consumer and Veterinary Affairs (legal authorizations: VD, FR and BE 3213 and 3571; capture and ringing permissions from the Federal Office for the Environment). The 5% weight limit of the GPS-ACC loggers was considered acceptable because of the short period during which the tracking devices were deployed on birds. Tagged birds survived and reproduced with similar annual fitness output than non-tagged individuals.

Reviewer #1 (Public Review): https://doi.org/10.7554/eLife.87775.3.sa1
Author response https://doi.org/10.7554/eLife.87775.3.sa2

# Additional files

## Supplementary files
• MDAR checklist

## Data availability
All code and datasets used to produce the results, analyses, and figures presented in this manuscript are available from a GitHub repository at https://github.com/kimschalcher/data-availability-Schalcher-et-al-eLife, copy archived at *Schalcher, 2024*.

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

**Appendix 1**

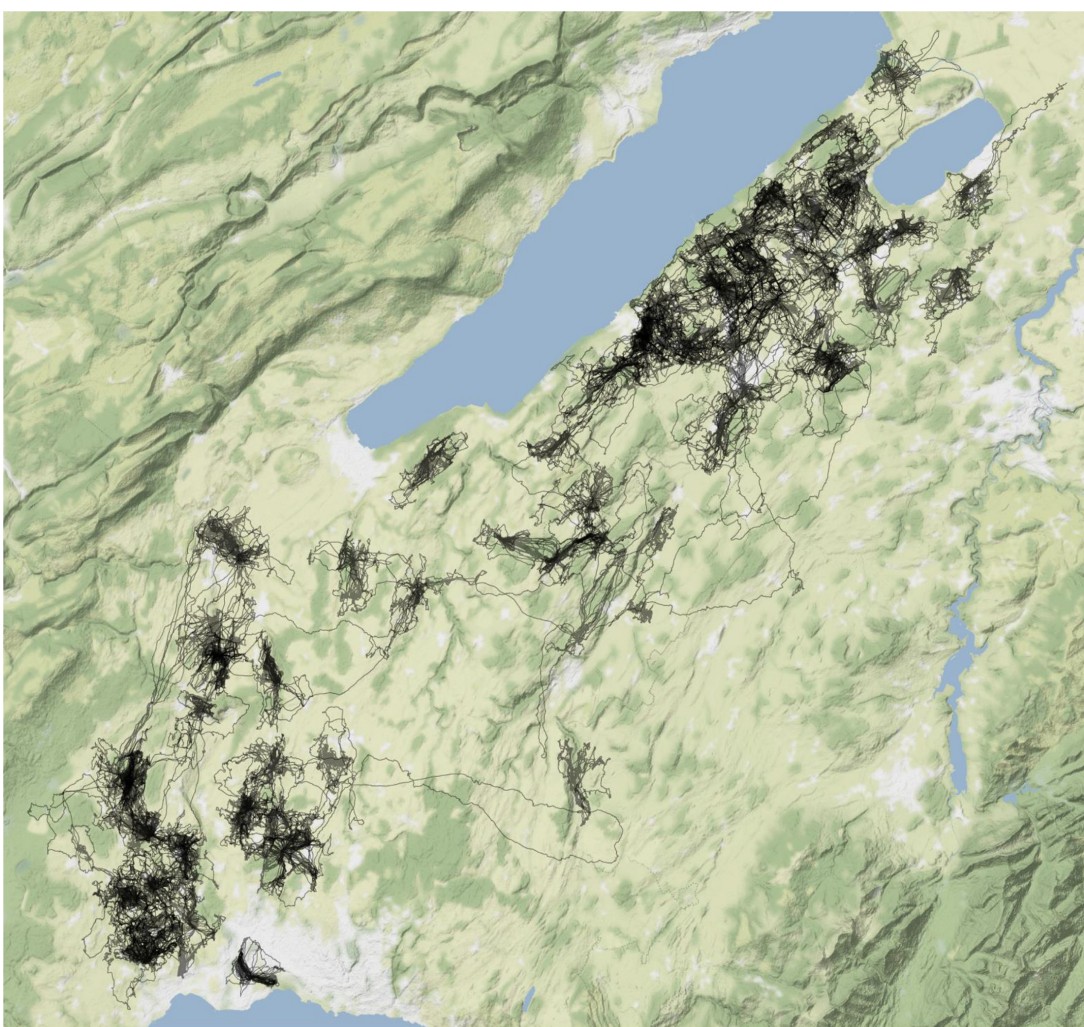

**Appendix 1—figure 1.** GPS tracks (in black) of the 163 breeding barn owls used in this study were recorded in 2019 and 2020 in western Switzerland.

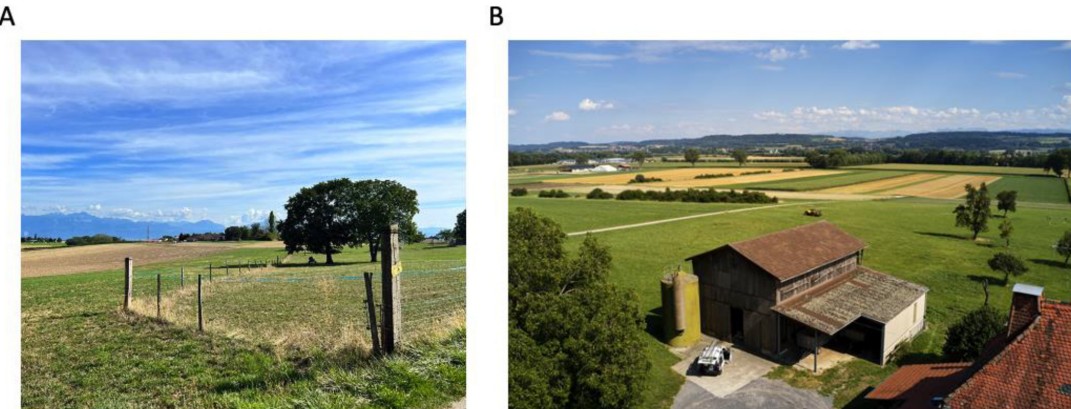

**Appendix 1—figure 2.** Typical barn owl foraging ground and nest location in western Switzerland. (**A**) Example of how pasture poles are usually located within the agricultural landscape which represents the main habitat for barn owls in western Switzerland. (**B**) Typical barn in which nest boxes are usually installed on the Swiss plateau.

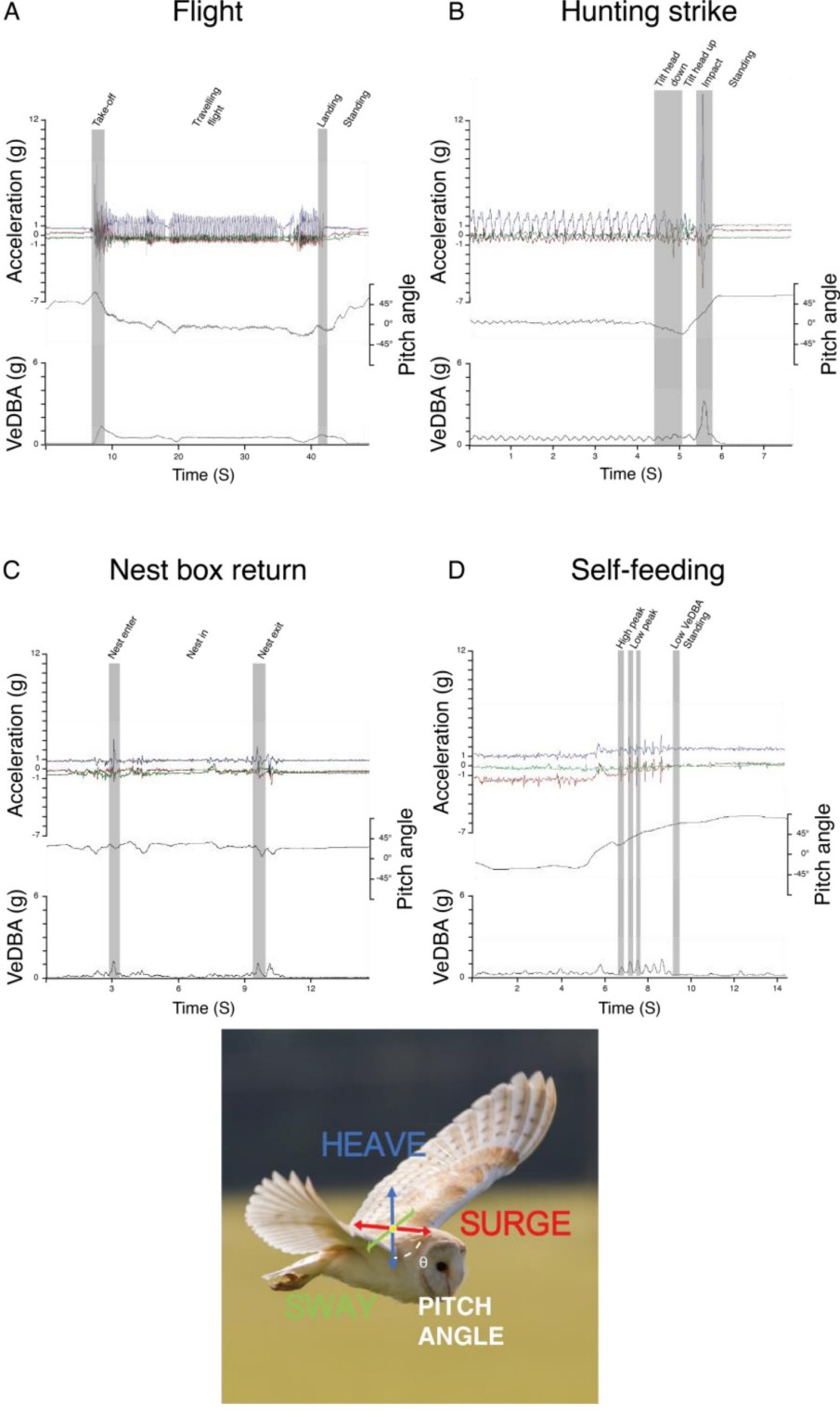

**Appendix 1—figure 3.** Behaviour classification from accelerometer data. Time series data of the different behaviour classified using Boolean approach showing changes in the raw tri-axial acceleration, body pitch angle, and the vectorial sum of the dynamic body acceleration (VeDBA) corresponding to (**A**) flight, (**B**) hunting strikes, (**C**) nest box visit, and (**D**) self-feeding events. Behaviour-specific base element used in the Boolean classification are shown in grey bands. Note that hunting strikes involves greater acceleration amplitude, VeDBA and body pitch variation between landings in context of usual perching (here at the end of flight sequence).

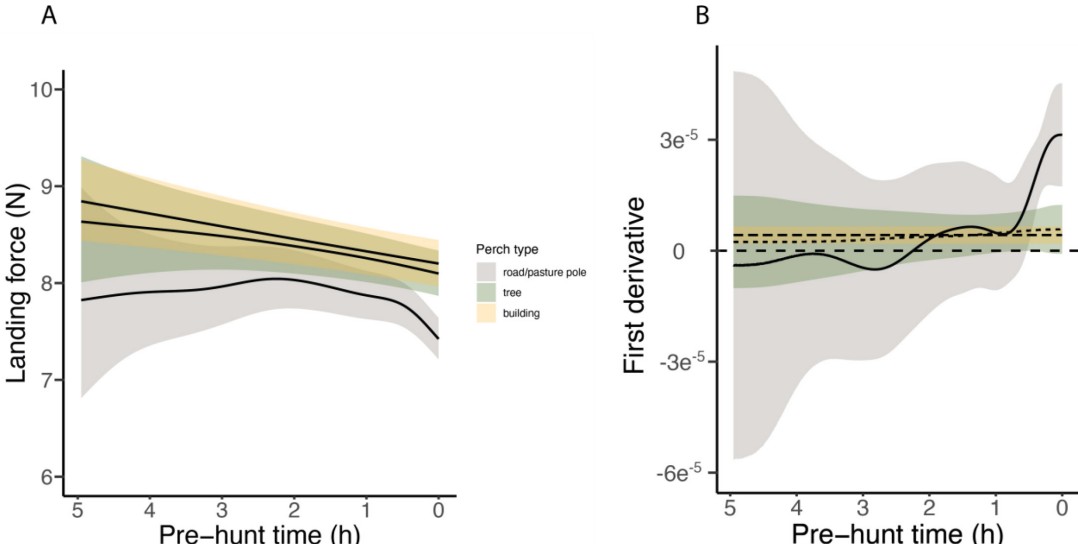

**Appendix 1—figure 4.** Complete variation in landing force according to pre-hunt time. Graphic representation of (**A**) the complete variation of the landing force calculated during perching and (**B**) the corresponding first derivative events in relation to the time (hours) until the next hunting event depending on whether owls perched on poles (in grey), trees (in green) and buildings (in yellow). Each line represents the predicted means for each perch type (averaged over male individuals), and shades show the 95% confidence intervals around each mean.

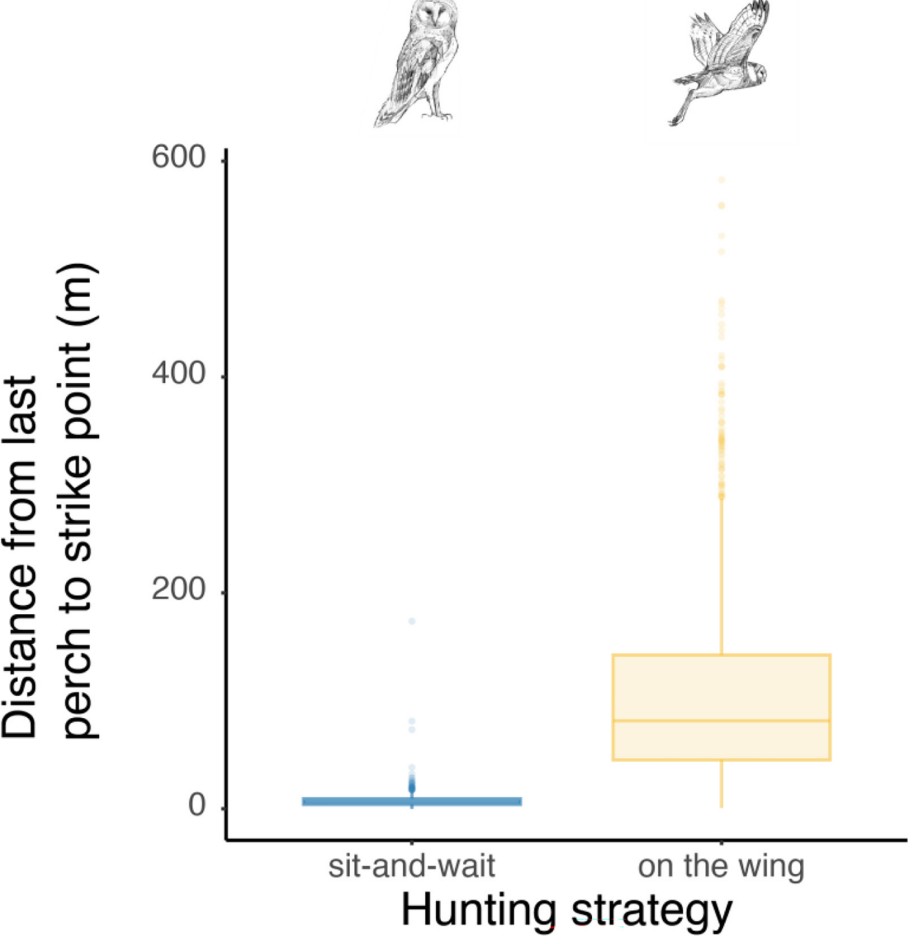

**Appendix 1—figure 5.** Differences in distance between the last perch and strike location according to the hunting strategy. Box plots of the variation in distance between perching location and hunting strikes among both perching and flying hunting strategies, showing a significant difference in distance between perch and strike location according to the hunting strategy (Wilcoxon test: W=2344881, p-value <0.001). Boxes boundaries highlight the first and the third quartile of the range distribution of the data. The line within each box marks the median and whiskers above and below boxes indicate the 10th and 90th percentiles. Owl drawings are courtesy of L. Willenegger, all used with permission.

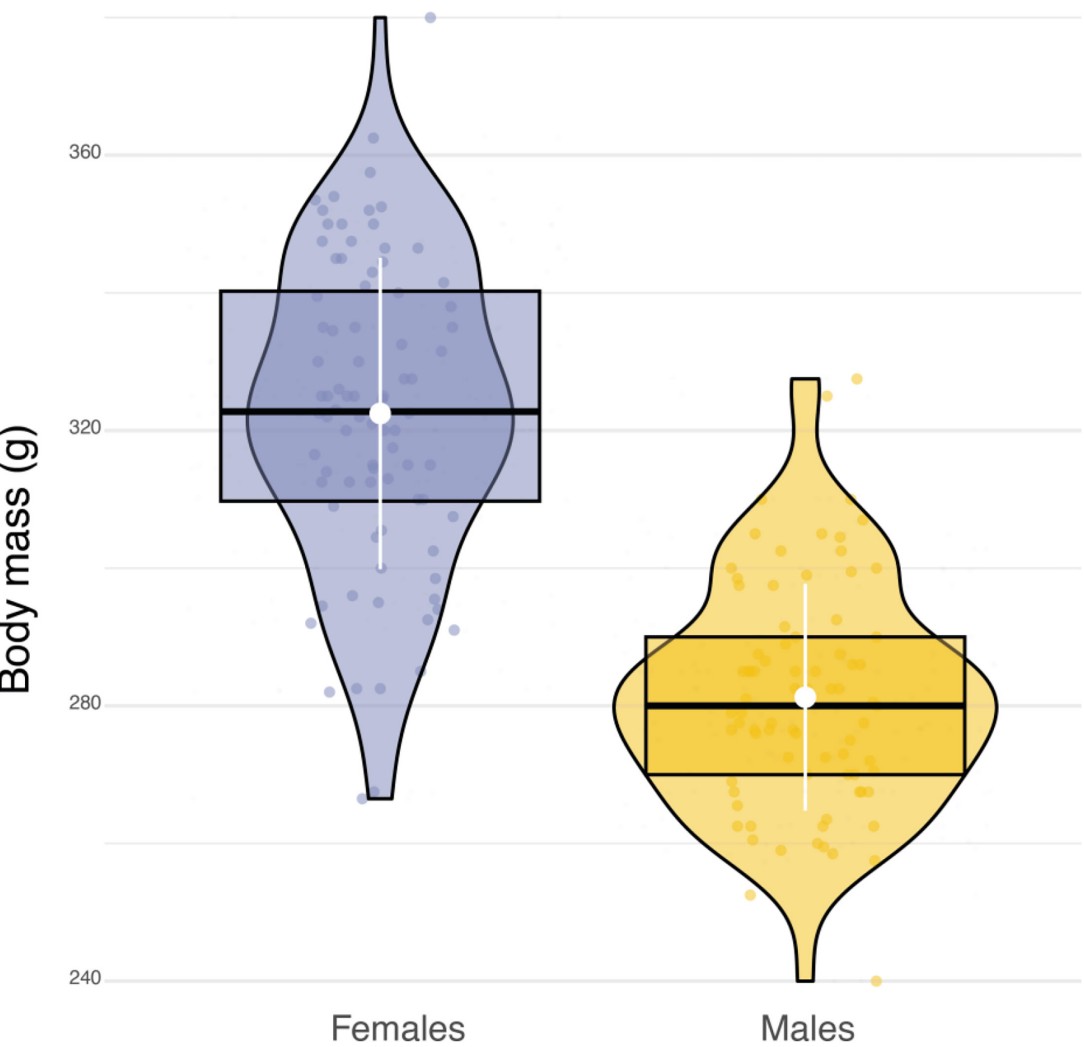

**Appendix 1—figure 6.** Sexual dimorphism in body mass. Box plots of the variation in body mass between females (blue dots, n=84) and males (orange dots, n=79). White dots and bars, respectively highlight the average and the standard deviation of body mass.

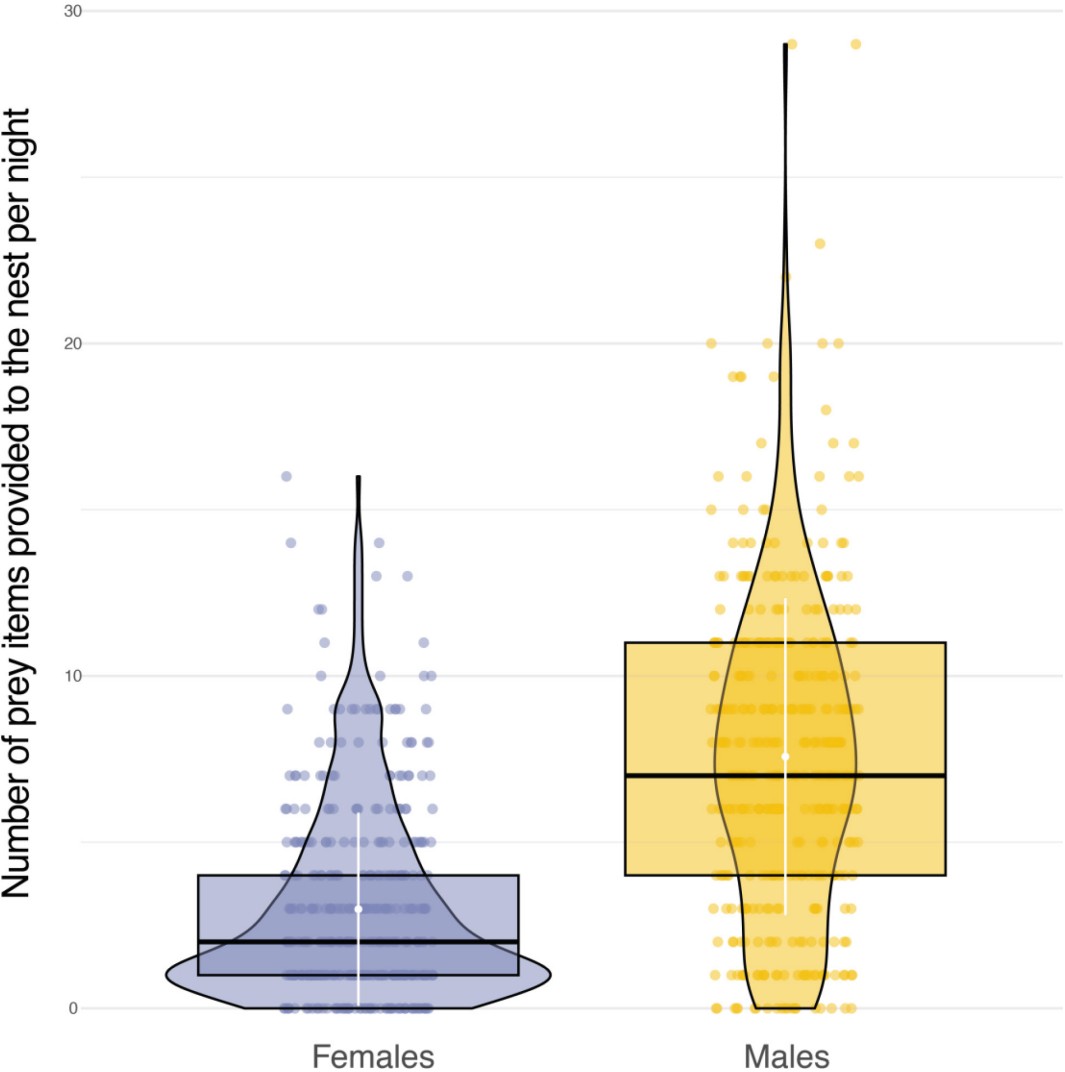

**Appendix 1—figure 7.** Sexual differences in food provisioning. Box plots of the variation in the number of prey items delivered to the nest each night between females (blue dots, n=1226) and males (orange dots, n=3105). White dots and bars, respectively highlight the average and the standard deviation of the number of prey items delivered to the nest each night.

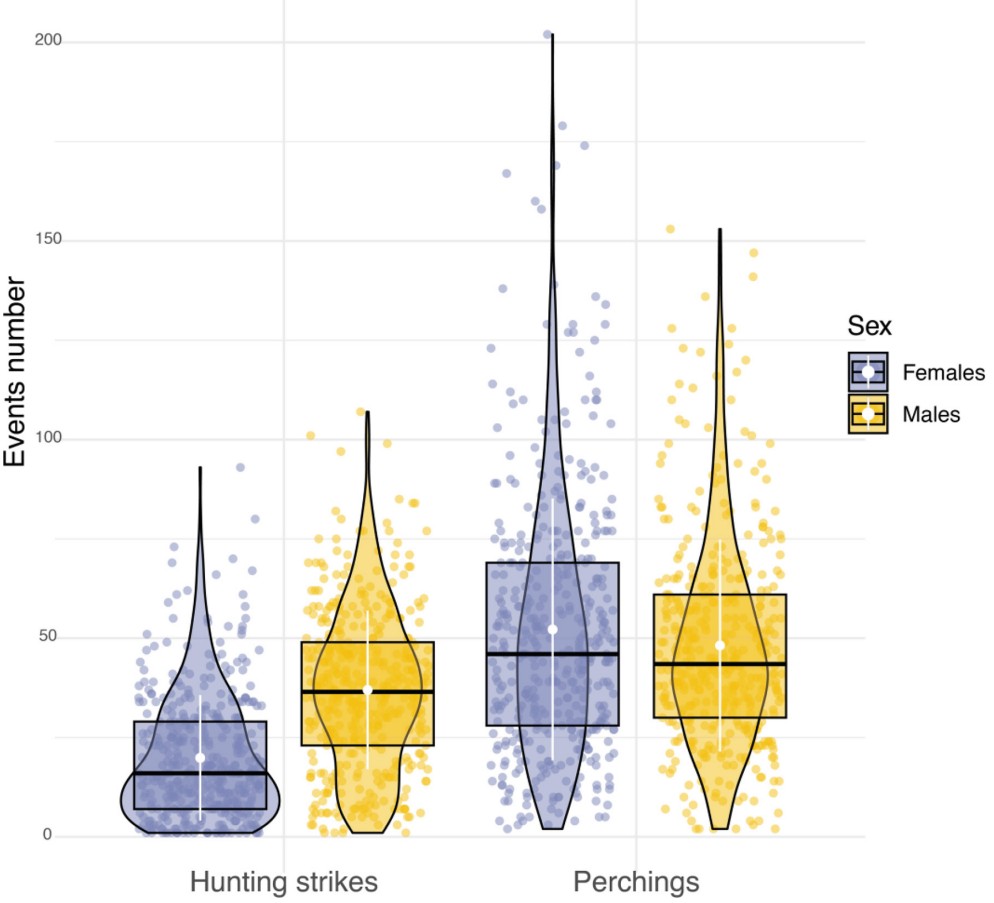

**Appendix 1—figure 8.** Sexual differences in hunting activity. Box plots of the variation in the number of hunting strikes and perching events performed each night by females (blue dots, nb perching events = 22,134, nb hunting strikes = 8176) and males (orange dots, nb perching events = 19,657, nb hunting strikes = 15,158). White dots and bars respectively highlight the average, and the standard deviation of the number of hunting strikes and perching events performed each night.

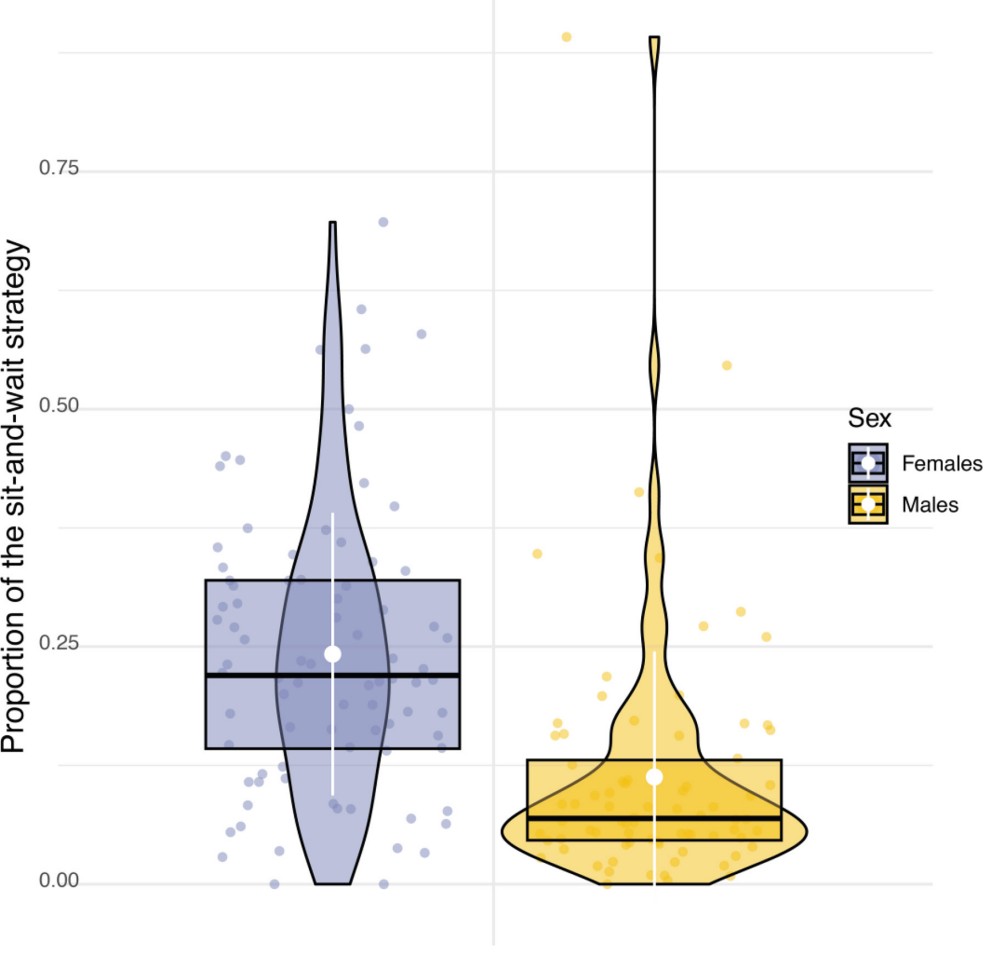

**Appendix 1—figure 9.** Sexual difference in the use of hunting strategy. Box plots of the variation in the proportion of use of the sit-and-wait strategy for females (blue dots) and males (orange dots). White dots and bars, respectively highlight the average and the standard deviation of the proportion of use of the sit-and-wait strategy.

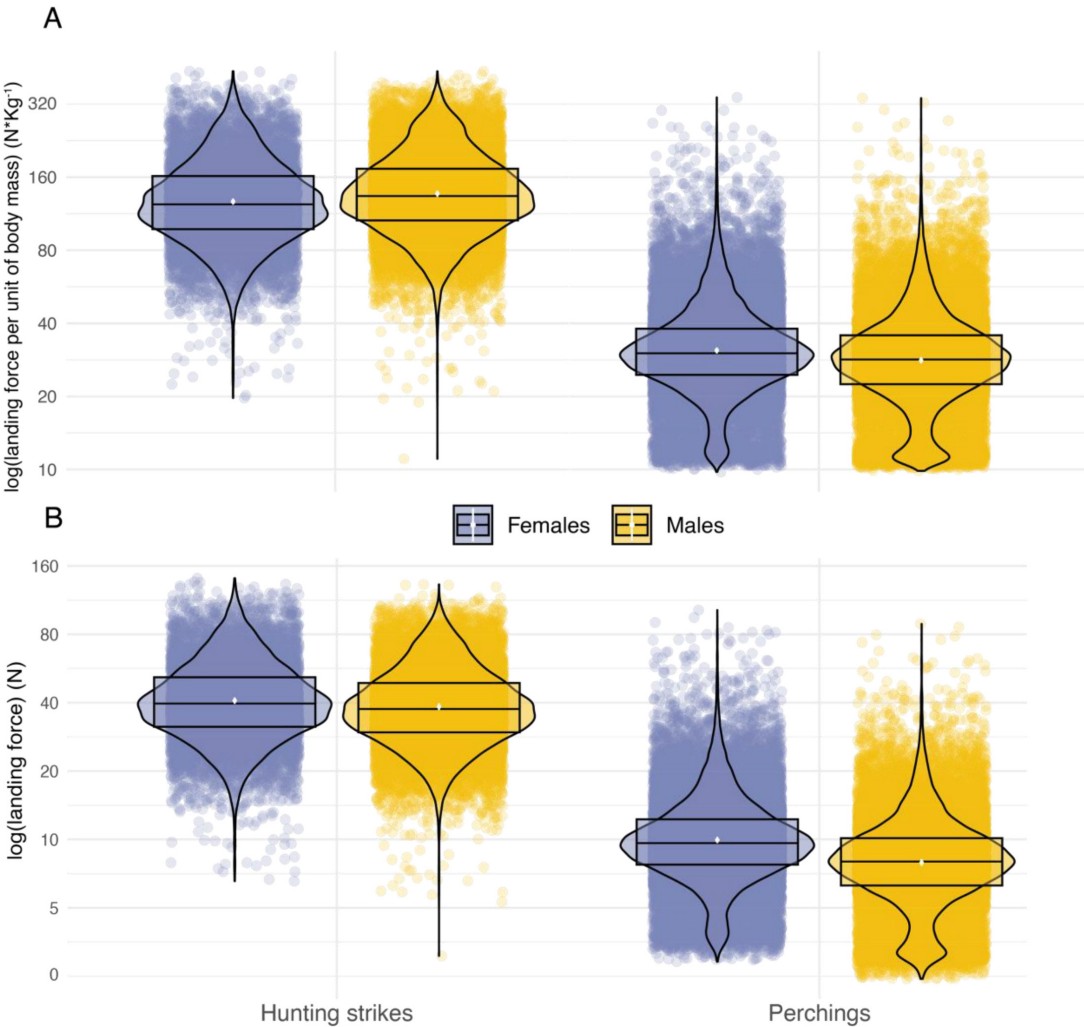

**Appendix 1—figure 10.** Sexual comparison of landing force as a function of landing context. Variation in peak landing force, on the log scale, involved in perching events and hunting strikes between female (blue dots, nb hunting strikes = 10,117, nb perching events = 30,378) and male (orange dots, nb hunting strikes = 17,864, nb perching events = 26,496) individuals. (**A**) Shows landing force per unit of body mass and (**B**) shows variation in peak landing force. White dots show the estimated mean, and data distribution is represented by both violin and box plots.

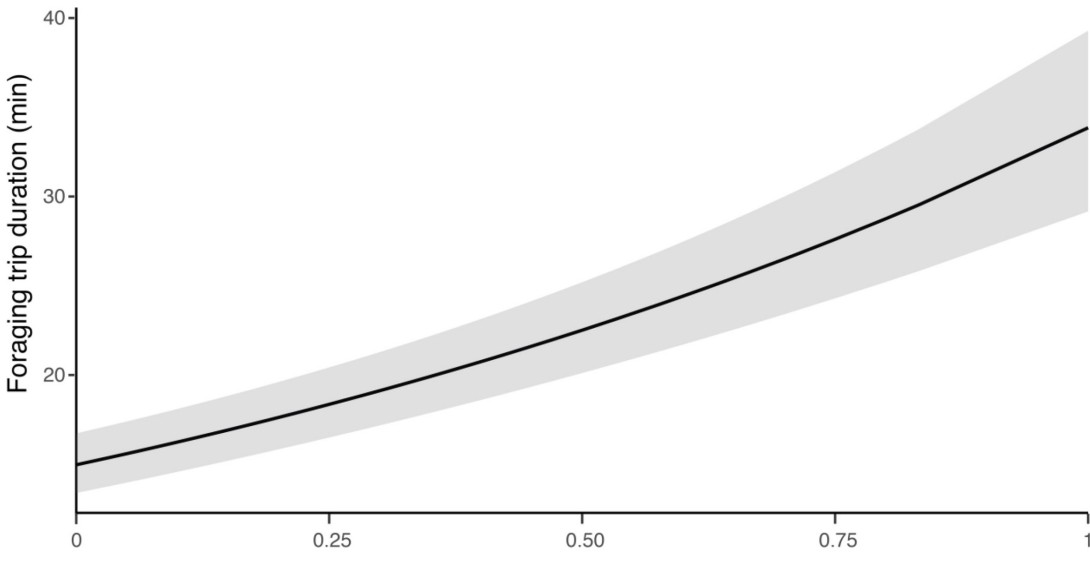

**Appendix 1—figure 11.** Influence of hunting strategy on barn owls foraging trip duration. Variation in foraging trip duration (in min) as a function of the frequency of use of the sit-and-wait strategy (relative to hunting on the wing). Solid line shows the estimated mean (averaged over both sexes) and the grey shade corresponds to the 95% confidence intervals around the mean.

**Appendix 1—table 1.** Time series (TS) of base elements applied for the classification of flight, hunting strikes, nest return (indicating prey delivery to the nest), and self-feeding behaviours. Each base element (BE) has a defined temporal flexibility that includes a number of events over which the conditions of the element are met (Present), as well as a range to the next element (Range), a window of flexibility (Flexibility) and a period over which the element is extended (ETNE), where the units are in events, 50 Hz having 50 events per second. See *Wilson et al., 2018* for details of use of these algorithms in the Boolean approach.

| Flight | BE | Present | Range | Flexibility | ETNE |
|---|---|---|---|---|---|
| TS1 | Take-off | 25 | 26 | 50 | |
| TS2 | Flying | 5 | 6 | 60'000 | 10 |
| TS3 | Landing | 90 | 91 | 5 | |
| TS4 | Standing | 4 | | | |

| Hunting strike | BE | Present | Range | Flexibility | ETNE |
|---|---|---|---|---|---|
| TS1 | Tilt head down | 5 | 20 | 40 | 10 |
| TS2 | Legs swinging | 3 | 5 | 110 | |
| TS3 | Impact | 1 | 1 | 160 | |
| TS4 | Standing | 5 | | | |

| Nest return | BE | Present | Range | Flexibility | ETNE |
|---|---|---|---|---|---|
| TS1 | Nest enter | 1 | 3 | 4 | |
| TS2 | Nest in | 1 | 50 | 1500 | 5 |
| TS3 | Nest exit | 1 | | | |

| Self feeding | BE | Present | Range | Flexibility | ETNE |
|---|---|---|---|---|---|
| TS1 | High peak | 1 | 1 | 15 | |

*Continued on next page*

*Continued*

| Self feeding | BE | Present | Range | Flexibility | ETNE |
|---|---|---|---|---|---|
| TS2 | Low peak | 1 | 1 | 15 | |
| TS3 | High peak | 1 | 1 | 15 | |
| TS4 | Low peak | 1 | 1 | 15 | |
| TS5 | Low VeDBA | 10 | 10 | 150 | |
| TS6 | End | 15 | | | |

**Appendix 1—table 2.** Model output for the linear mixed model (LMM) predicting variations in landing force (log-transformed) due to landing context (hunting strike vs perching) and sex (Males vs Females).

The model also included a random effect of BirdID and NightID (nested in BirdID). Intercept is reported for both landing contexts (highlighted in grey) and give information about the averaged landing force considering female individuals. Estimates for interactions give the % of change between females and males for each landing contexts. Variance ($\sigma^2$), intra-class correlation coefficient (ICC), and number of observations are provided for random effects.

| | Impact force (N) | | |
|---|---|---|---|
| *Predictors* | *Estimate* | *Conf. Int.* | *p-value* |
| Landing context [strike] | 40.81 | 39.49–42.18 | <0.001 |
| Landing context [perching] | 9.94 | 9.63–10.27 | <0.001 |
| Landing context [strike] * sex[M] | 0.94 | 0.90–0.99 | 0.011 |
| Landing context [perching] * sex[M] | 0.80 | 0.76–0.83 | <0.001 |
| Observations | 84855 | | |
| Marginal R² /Conditional R² | 0.736/0.772 | | |

Random Effects: $\sigma^2$=0.15 | ICC = 0.14 | $N_{BirdID}$ = 163 | $N_{NightID}$ = 7

**Appendix 1—table 3.** Model selection results using Akaike Information Criteria corrected for small sample sizes (AICc) for all possible models evaluating the effect of landing context and sex on barn owl landing force.

A '*' in the variable columns indicates that the variable was included in that model. K is the number of variables included in each model. All linear mixed model (LMM) included a random effect of BirdID and NightID (nested in BirdID). Table includes all possible models, ranked by AICc. Models with Delta AICc <2 are highlighted in grey and the final model is shown in bold.

| # | Sex | Landing context | Sex: Landing context | K | AIC | ΔAIC | Model weight |
|---|---|---|---|---|---|---|---|
| **1** | * | * | * | **3** | **82517** | **0** | **1** |
| 2 | * | * | | 2 | 83252.5 | 735.51 | 0 |
| 3 | * | | | 1 | 83297.9 | 780.84 | 0 |
| 4 | | * | | 1 | 198495.8 | 115978.81 | 0 |
| 5 | | | | 0 | 198498.4 | 115981.36 | 0 |

**Appendix 1—table 4.** Model output of the generalized additive mixed-effects model (GAMM) predicting variations in pre-hunt perching force in N (log-transformed) due to perch type (buildings, tree branches, and road/pasture poles), wind speed, and sex as linear predictor (lme) and time to the next strike as an additive term (gam).

The model also included a random effect of BirdID. Effective degrees of freedom (EDF) are shown for additive terms, providing the degree of non-linearity between pre-hunt perching force and time to hunt for each perch type.

| LME | Impact force (N) | | |
|---|---|---|---|
| *Predictors* | *Estimate* | *Conf. Int.* | *p-value* |
| Intercept | 8.99 | 8.74–9.24 | <0.001 |
| Perch type [tree] | 1.07 | 1.07–1.08 | <0.001 |
| Perch type [buildings] | 1.09 | 1.08–1.09 | <0.001 |
| Wind speed | 1.00 | 0.99–1.00 | 0.04 |
| Sex [Male] | 0.85 | 0.81–0.88 | <0.001 |
| GAM | | | |
| *Predictors* | *EDF* | *F* | *p-value* |
| Time to hunt: perch type [road/pasture] | 4.22 | 22.43 | <0.001 |
| Time to hunt: perch type [tree] | 1.50 | 5.12 | 0.005 |
| Time to hunt: perch type [buildings] | 1.00 | 12.86 | <0.001 |
| Observations | 27981 | | |
| $R^2$ | 0.235 | | |

**Appendix 1—table 5.** Model selection results using Akaike Information Criteria corrected for small sample sizes (AICc) for all possible models evaluating the effect of time to hunt and perch type on landing force during perching events.

A '*' in the variable columns indicates that the variable was included in that model. K is the number of variables included in each model. All linear mixed model (LMM) included a random effect of BirdID and NightID (nested in BirdID). Table includes the top five models, ranked by AICc. Models with Delta AICc <2 are highlighted in grey and the final model is shown in bold.

| # | Perch type | Wind speed | Sex | s (time to hunt: perch type) | K | AICc | ΔAICc | Model weight |
|---|---|---|---|---|---|---|---|---|
| 1 | * | * | * | * | 4 | 203.1 | 0 | 0.48 |
| 2 | * | * | | * | 3 | 204.5 | 1.42 | 0.24 |
| 3 | * | | * | * | 3 | 205.0 | 1.92 | 0.18 |
| 4 | * | | | * | 2 | 206.4 | 3.26 | 0.09 |
| 5 | * | * | * | | 3 | 324.2 | 121.13 | 0 |

**Appendix 1—table 6.** Model output for the linear mixed model (LMM) predicting variations in hunting strike force (log-transformed) due to hunting success (0 vs 1), hunting strategy (perching vs flying), and sex.

The model also included a random effect of BirdID and NightID (nested in BirdID). Intercept provides the averaged strike force (N) considering female individuals hunting on the wing. Variance ($\sigma^2$), intra-class correlation coefficient (ICC), and number of observations are provided for random effects.

| | Impact force (N) | | |
|---|---|---|---|
| *Predictors* | *Estimate* | *Conf. Int.* | *p-value* |
| Intercept | 41.51 | 40.31–42.75 | <0.001 |
| Hunting success [0] | 0.95 | 0.94–0.96 | <0.001 |
| Sex [M] | 0.94 | 0.91–0.98 | 0.005 |
| Hunting strategy [sit-and-wait] | 0.96 | 0.94–0.99 | 0.002 |
| Hunting success [0] * hunting strategy [sit-and-wait] | 1.04 | 1.01–1.07 | 0.004 |
| Observations | 27981 | | |

*Appendix 1—table 6 Continued on next page*

*Appendix 1—table 6 Continued*

|  | Impact force (N) |
| --- | --- |
| Marginal R² /Conditional R² | 0.007/0.115 |

Random Effects: o²=0.13 | ICC = 0.11 | N$_{BirdID}$ = 163 | N$_{NightID}$ = 7

**Appendix 1—table 7.** Model selection results using Akaike Information Criteria corrected for small sample sizes (AICc) for all possible models evaluating the effect of hunting success and hunting strategy on landing force during hunting strike.

A '*' in the variable columns indicates that the variable was included in that model. K is the number of variables included in each model. All linear mixed model (LMM) included a random effect of BirdID and NightID (nested in BirdID). Table includes the top five models, ranked by AICc. Models with Delta AICc <2 are highlighted in grey and the final model is shown in bold.

| # | Sex | Hunting success | Hunting strategy | Hunting strategy: Hunting success | K | AIC | ΔAIC | Model weight |
| --- | --- | --- | --- | --- | --- | --- | --- | --- |
| 1 | * | * | * | * | 4 | 23996.3 | 0 | 0.87 |
| 2 | * | * |  |  | 2 | 24002.1 | 5.82 | 0.05 |
| 3 |  | * | * | * | 3 | 24002.1 | 5.82 | 0.05 |
| 4 | * | * | * |  | 3 | 24002.7 | 6.36 | 0.04 |
| 5 |  | * |  |  | 1 | 24007.7 | 11.41 | 0.003 |

**Appendix 1—table 8.** Model output for the generalized linear mixed-effect model (GLMM) predicting variations in hunting success (binary response 0,1) due to sex and hunting strategy (on the wing vs sit-and-wait).

The model also included a random effect of BirdID and NightID (nested in BirdID). Intercept is reported for both sexes and give information about the averaged hunting success considering individuals hunting on the wing. Standardized estimates are provided for any additional terms in the model, representing % of change (odds ratio) of hunting success. The influence of landing force on hunting success is provided considering both hunting strategies. Variance (σ²), intra-class correlation coefficient (ICC), and number of observations of each group are provided for random effects.

|  | Hunting success | | |
| --- | --- | --- | --- |
| *Predictors* | *Estimate* | *Conf. Int.* | *p-value* |
| Sex [Female] | 0.24 | 0.22–0.26 | <0.001 |
| Sex [Male] | 0.35 | 0.33–0.38 | <0.001 |
| Hunting strategy [sit-and-wait]: Females | 1.55 | 1.38–1.75 | <0.001 |
| Hunting strategy [sit-and-wait]: Males | 1.49 | 1.32–1.68 | <0.001 |
| Observations | 27981 | | |
| Marginal R² /Conditional R² | 0.014/0.051 | | |

Random Effects: o²=3.29 | ICC = 0.04 | N$_{BirdID}$ = 163 | N$_{NightID}$ = 7

**Appendix 1—table 9.** Model selection results using Akaike Information Criteria corrected for small sample sizes (AICc) for all possible models evaluating the effect of sex, hunting strategy, and their interaction on hunting success.

A '*' in the variable columns indicates that the variable was included in that model. K is the number of variables included in each model. All linear mixed model (LMM) included a random effect of BirdID and NightID (nested in BirdID). Table includes the top five models, ranked by AICc. Models with Delta AICc <2 are highlighted in grey and the final model is shown in bold.

| # | Sex | Hunting strategy | Sex:Hunting strategy | K | AIC | ΔAIC | Model weight |
|---|-----|------------------|----------------------|---|-----|------|--------------|
| 1 | * | * | | 2 | 30726.8 | 0 | 0.71 |
| 2 | * | * | * | 3 | 30728.6 | 1.79 | 0.29 |
| 3 | * | | | 1 | 30768.0 | 41.22 | 0 |
| 4 | | * | | 1 | 30816.2 | 89.38 | 0 |
| 5 | | | | 0 | 30848.5 | 121.68 | 0 |

**Appendix 1—table 10.** Modelling the effect of landing force during pre-hunt perching on hunting success.

Model output for the generalized linear mixed-effect model (GLMM) predicting variations in hunting success (binary response 0,1) due to pre-hunt perching force, sex, hunting strategy (on the wing vs sit-and-wait), and wind speed. The model also included a random effect of BirdID and NightID (nested in BirdID). Intercept is reported for both sexes (highlighted in grey) and give information about the average hunting success considering individuals hunting on the wing. Standardized estimates are provided for any additional terms in the model, representing % of change (odds ratio) of hunting success. The influence of landing force on hunting success is provided considering both hunting strategies. Variance ($\sigma^2$), intra-class correlation coefficient (ICC), and number of observations of each group are provided for random effects.

| | Hunting success | | |
|---|---|---|---|
| *Predictors* | *Estimate* | *Conf. Int.* | *p-value* |
| Females | 0.22 | 0.18–0.26 | <0.001 |
| Males | 0.32 | 0.28–0.37 | <0.001 |
| Hunting strategy [sit-and-wait] | 1.51 | 1.26–1.81 | <0.001 |
| Hunting strategy [on the wing]: Pre-hunt perching force | 1.08 | 0.97–1.20 | 0.178 |
| Hunting strategy [sit-and-wait]: Pre-hunt perching force | 0.85 | 0.73–0.99 | 0.037 |
| Observations | 3040 | | |
| Marginal $R^2$ /Conditional $R^2$ | 0.023/0.043 | | |

Random Effects: $\sigma^2$=3.29 | ICC = 0.02 | $N_{BirdID}$ = 151 | $N_{NightID}$ = 7

**Appendix 1—table 11.** Model selection results using Akaike Information Criteria corrected for small sample sizes (AICc) for all possible models evaluating the effect of landing force during pre-hunt perching on hunting success.

A '*' in the variable columns indicates that the variable was included in that model. K is the number of variables included in each model. All linear mixed model (LMM) included a random effect of BirdID and NightID (nested in BirdID). Table includes the top five models, ranked by AICc. Models with Delta AICc <2 are highlighted in grey and the final model is shown in bold.

| # | Sex | Wind speed | Perch type | Hunting strategy | Pre-hunt perching force | Hunting strategy: Pre-hunt perching force | K | AIC | ΔAIC | Model weight |
|---|-----|------------|------------|------------------|-------------------------|-------------------------------------------|---|-----|------|--------------|
| 1 | * | | | * | * | * | 4 | 3326.5 | 0 | 0.24 |
| 2 | * | | * | * | * | * | 5 | 3327.1 | 0.54 | 0.19 |
| 3 | * | * | | * | * | * | 5 | 3327.4 | 0.82 | 0.16 |
| 4 | * | * | * | * | * | * | 6 | 3328.1 | 1.55 | 0.11 |
| 5 | * | | | * | | | 2 | 3329.1 | 2.56 | 0.07 |

**Appendix 1—table 12.** Model output of the linear mixed model (LMM) predicting variation of barn owls foraging flights speed (in ms$^{-1}$) as a function of the sex.

Intercept provides the averaged foraging flight speed (in ms$^{-1}$) considering female individuals.

The model also included a random effect of BirdID and NightID (nested in BirdID). Variance ($\sigma^2$), intra-class correlation coefficient (ICC), and number of observations of each group are provided for random effects.

| Predictors | Foraging flight speed (ms$^{-1}$) | | |
| --- | --- | --- | --- |
| | Estimate | Conf. Int. | p-value |
| Intercept | 5.47 | 5.38–5.56 | <0.001 |
| Sex [Male] | –0.23 | –0.36 to –0.10 | <0.001 |
| Observations | 27242 | | |
| Marginal R$^2$ /Conditional R$^2$ | 0.006/0.141 | | |

Random Effects: $\sigma^2$=1.60 | ICC = 0.14 | N$_{BirdID}$ = 163 | N$_{NightID}$ = 7

**Appendix 1—table 13.** Model output of the linear mixed model (LMM) predicting variation of barn owls foraging trip duration (in min) as a function of the total number of hunting attempts per trip, the frequency of use of the sit-and-wait strategy (relative to hunting on the wing), and sex.

The model also included a random effect of BirdID and NightID (nested in BirdID). Variance ($\sigma^2$), intra-class correlation coefficient (ICC), and number of observations of each group are provided for random effects.

| Predictors | Foraging trip duration (min) | | |
| --- | --- | --- | --- |
| | Estimate | Conf. Int. | p-value |
| Intercept | 9.49 | 8.43–10.67 | <0.001 |
| Nb of hunting attempts | 1.16 | 1.16–1.17 | <0.001 |
| Freq sit-and-wait | 2.26 | 1.97–2.59 | <0.001 |
| Sex [Male] | 0.79 | 0.69–0.90 | 0.001 |
| Observations | 27242 | | |
| Marginal R$^2$ /Conditional R$^2$ | 0.006/0.141 | | |

Random Effects: $\sigma^2$=1 | ICC = 0.15 | N$_{BirdID}$ = 150 | N$_{NightID}$ = 7

