## [Editor Report · eLife assessment]

This **fundamental** work substantially advances our understanding of animals' foraging behaviour by monitoring the movement and body posture of barn owls in high resolution and assessing their foraging success. With a large dataset, the evidence supporting the main conclusions is **compelling**. This work provides new corroboration for motion-induced sound camouflage and has broad implications for understanding predator-prey interactions.

---

## [Referee Report · Reviewer #1 (Public Review)]

In this paper, Schalcher et al. examined how barn owls' landing force affects their hunting success during two hunting strategies: strike hunting and sit-and-wait hunting. They tracked tens of barn owls that raised their nestlings in nest boxes and utilized high-resolution GPS and acceleration loggers to monitor their movement. In addition, camcorders were placed near their nest boxes and used to record the prey they brought to the nest, thus measuring their foraging success.

This study generated a unique dataset and provided new insights into the foraging behavior of barn owls. The researchers discovered that the landing force during hunting strikes was significantly higher compared to the sit-and-wait strategy. Additionally, they found a positive relationship between landing force and foraging success during hunting strikes, whereas, during the sit-and-wait strategy, there was a negative relationship between the two. This suggests that barn owls avoid detection by generating a lower landing force and producing less noise. Furthermore, the researchers observed that environmental characteristics affect barn owls' landing force during sit-and-wait hunting. They found a greater landing force when landing on buildings, a lower landing force when landing on trees, and the lowest landing force when landing on poles. The landing force also decreased as the time to the next hunting attempt decreased. These findings collectively suggest that barn owls reduce their landing force as an acoustic camouflage to avoid detection by their prey.

The main strength of this work is the researchers' comprehensive approach, examining different aspects of foraging behavior, including high-resolution movement, foraging success, and the influence of the environment on this behavior, supported by impressive data collection.

The results presented support the authors' conclusion that lower landing force during sit-and-wait hunting increases hunting success, likely due to a decreased probability of detection by their prey, resulting in acoustic camouflage. The authors also hypothesized that hunting success is crucial for survival, and thus, acoustic camouflage has a direct link to fitness. This paper provides an unprecedented dataset and the first measurement of landing force during hunting in the wild. It is likely to inspire many other researchers currently studying animal foraging behavior to explore how animals' movement affects foraging success.

---

## [Author Response]

We would like to thank you and the reviewers for your thoughtful comments that assisted us to improve the manuscript. We carefully followed the reviewers’ recommendations and provide a detailed point-by-point account of our responses to the comments.

Please find below the important changes in the updated manuscript.

(1) We changed the title according to the comments provided by reviewer #1.

(2) We edited the introduction, results, and discussion to improve the link between the objectives of the study, the findings, and their discussion, as reviewer #2 recommended.

(3) We clarified the link between camouflage and fitness, which is now presented as a hypothesis, as reviewer #1 suggested.

(4) We added new analyses and figures in the main text and in the supplementary materials to better emphasize sex differences in landing force, foraging strategies and hunting success, following reviewer #1 suggestion.

(5) According to reviewer #2 comments, we edited the results adding key information about methods to help the reader understand the findings without reading the Methods section.

(6) We added important details about the model selection approach along with a discussion of the low R-square values reported in our analyses on hunting success, as reviewer #2 suggested.

**eLife assessment**

This fundamental work substantially advances our understanding of animals' foraging behaviour, by monitoring the movement and body posture of barn owls in high resolution, in addition to assessing their foraging success. With a large dataset, the evidence supporting the main conclusions is convincing. This work provides new evidence for motion-induced sound camouflage and has broad implications for understanding predator-prey interactions.

**Public Reviews:**

**Reviewer #1 (Public Review):**
In this paper, Schalcher et al. examined how barn owls' landing force affects their hunting success during two hunting strategies: strike hunting and sit-and-wait hunting. They tracked tens of barn owls that raised their nestlings in nest boxes and utilized high-resolution GPS and acceleration loggers to monitor their movements. In addition, camcorders were placed near their nest boxes and used to record the prey they brought to the nest, thus measuring their foraging success.This study generated a unique dataset and provided new insights into the foraging behavior of barn owls. The researchers discovered that the landing force during hunting strikes was significantly higher compared to the sit-and-wait strategy. Additionally, they found a positive relationship between landing force and foraging success during hunting strikes, whereas, during the sit-and-wait strategy, there was a negative relationship between the two. This suggests that barn owls avoid detection by generating a lower landing force and producing less noise. Furthermore, the researchers observed that environmental characteristics affect barn owls' landing force during sit-and-wait hunting. They found a greater landing force when landing on buildings, a lower landing force when landing on trees, and the lowest landing force when landing on poles. The landing force also decreased as the time to the next hunting attempt decreased. These findings collectively suggest that barn owls reduce their landing force as an acoustic camouflage to avoid detection by their prey.The main strength of this work is the researchers' comprehensive approach, examining different aspects of foraging behavior, including high-resolution movement, foraging success, and the influence of the environment on this behavior, supported by impressive data collection. The weakness of this study is that the results only present a partial biological story contained within the data. The focus is on acoustic camouflage without addressing other aspects of barn owls' foraging strategy, leaving the reader with many unanswered questions. These include individual differences, direct measurements of owls' fitness, a detailed analysis of the foraging strategy of males and females, and the collective effort per nest box. However, it is possible that these data will be published in a separate paper.

We greatly appreciate your recognition of the comprehensive approach and extensive data collection. Our primary objective was to study the role of acoustic camouflage. Nonetheless, the manuscript now includes a detailed analysis of the foraging strategy and hunting success of males and females (lines 164-225).

The results presented support the authors' conclusion that lower landing force during sit-andwait hunting increases hunting success, likely due to a decreased probability of detection by their prey, resulting in acoustic camouflage. The authors also argue that hunting success is crucial for survival, and thus, acoustic camouflage has a direct link to fitness. While this statement is reasonable, it should be presented as a hypothesis, as no direct evidence has been provided here.

Thank you for the comment. We agree and thus have edited the language accordingly.

However, since information about nestling survival is typically monitored when studying behavior during the breeding period, the authors' knowledge of the effect of acoustic camouflage on owls' fitness can probably be provided. Furthermore, it will be interesting to further examine the foraging strategies used by different individuals during foraging, the joint foraging success of both males and females within each nest box, and the link between landing force and foraging success if the data are available.

We are currently writing a manuscript on these topics. We are aware that several scientific questions regarding the foraging ecology of the barn owl still need our attention. Regarding the link between landing force and foraging success, we believe that our revised manuscript addresses this specific topic, please see specific responses below.

However, even without this additional analysis on survival, this paper provides an unprecedented dataset and the first measurement of landing force during hunting in the wild. It is likely to inspire many other researchers currently studying animal foraging behavior to explore how animals' movements affect foraging success.Reviewer #2 (Public Review):Summary:The authors provide new evidence for motion-induced sound camouflage and can link the hunting approach to hunting success (detailing the adaptation and inferring a fitness consequence).Strengths:Strong evidence by combining high-resolution accelerometer data with a ground-truthed data set on prey provisioning at nest boxes. A good set of co-variates to control for some of the noise in the data provides some additional insights into owl hunting attempts.Weaknesses:There is a disconnect between the hypotheses tested and the results presented, and insufficient detail is provided on the statistical approach. R2 values of the presented models are very small compared to the significance of the effect presented. Without more detail, it is impossible to assess the strength of the evidence.

In the revised manuscript, we changed the way results are presented and we improved the link between the hypotheses and the results. The R2 values are indeed small. It is however important to keep in mind that we are assessing the outcome of one specific behavior (i.e. landing force during sit-and-wait hunts) on hunting success in a wild environment, where many complex ecological interactions likely influence hunting success. Nonetheless, the coefficients (as reported in the results) show that for every 1 N increase in landing force, there is a 15% reduction in hunting success, which is substantial. In the discussion we also note that 50 Hz is a relatively low sampling frequency for estimating the peak ground reaction force. We have gone back over the presentation of our results and made our discussion more nuanced to acknowledge this aspect.

We have also added a detailed description about our model selection process in the methods section and provide a model selection table for each analysis in the supplementary materials.

The authors seem to overcome persisting challenges associated with the validation and calibration of accelerometer data by ground-truthing on-board measures with direct observations in captivity, but here the methods are not described any further and sample sizes (2 owls - how many different loggers were deployed?) might be too small to achieve robust behavioural classifications.

Thank you for the comment. Details of our methods of behavioural identification are provided in lines 385 – 429. There are two reasons why our results should not be limited by the sample size. First, we used the temporal sequence of changes in acceleration, and rates of change in acceleration data, which make the methods robust to individual differences in acceleration values. Furthermore, our methods for behavioural identification were not based on machine learning. Instead, we use a Boolean based approach (as described in Wilson et al. 2018. MEE), which is more robust to small differences in absolute values that might occur e.g. in relation to slight changes in device position.

**Recommendation for the authors:**

**Reviewer #1 (Recommendations For The Authors):**
Comment 1. This study provides new insights into animals' foraging behavior and will probably inspire other researchers to examine foraging behavior in such high resolution.

We hope so, thank you.

Comment 2. However, it is necessary to describe better the measured landing force and the hunting strike and perching behavior so the readers can understand these methods when reading the results (and without reading the Methods).

We have now changed the text in the “Results” to help the reader understand the key methods while reading the results.

Comment 3. In addition, make sure you use the same terminology for hunting strategies during the entire paper and especially in all figures and corresponding result descriptions.

We now use consistent terminology throughout the text and figures. We hope that this is now clear in the revised manuscript.

Comment 4. In addition, although I find your statement about the link between acoustic camouflage and fitness reasonable, it should be described as a hypothesis or examined if you want to keep the direct link statement. I believe showing a direct link can add an additional outstanding aspect to this paper, but I also understand that it can be addressed in a separate paper.

We agree that the relationship between hunting success and barn owl fitness is an important topic, but it necessitates a consideration of both hunting strategies, including hunting on the wing, which extends beyond the limits of our current study. Indeed, our primary objective was to conduct a detailed examination of the interplay between acoustic camouflage and the success of the sit-and-wait technique.

However, we have edited the manuscript to explicitly describe the link between acoustic camouflage and fitness as a hypothesis. We believe this adjustment provides a more accurate representation of our approach. We hope this clarifies the specific emphasis of our work and its contribution to the understanding of barn owl hunting behavior.

Here are my detailed comments about the paper:Comment 5. Title: Consider changing the title to "Acoustic camouflage predicts hunting success in a wild predator."

We would like to thank you for your nice proposition. However, we opted for a different title, which is now “Landing force reveals new form of motion-induced sound camouflage in a wild predator”.

Comment 6. Line 91-93: Please provide additional information about the collected dataset, including:Description of the total period of observations, an average and standard deviation of perching and hunting attempt events per individual per night, number of foraging trips per individual per night, details about the geographic location and characteristics of the habitat, season, and reproductive state.

The revised manuscript now includes detailed information about the collected dataset (i.e. study area, reproductive state, etc…). “We used GPS loggers and accelerometers to record high resolution movement data during two consecutive breeding seasons (May to August in 2019 and 2020) from 163 wild barn owls (79 males and 84 females) breeding in nest boxes across a 1,000 km² intensive agricultural landscape in the western Swiss plateau.” Results section, lines 79 – 82

Details about the number of foraging trips per individuals and per night are now presented in the results: “Sexual dimorphism in body mass was marked among our sampled individuals. Males were lighter than females (84 females, average body mass: 322 ± 22.6 g; 79 males, average body mass 281 ± 16.5 g, Fig S6) and provided almost three times more prey per night than females (males: 8 ± 5 prey per night; females: 3 ± 3 prey per night; Fig.S7). Males also displayed higher nightly hunting effort than females (Males: 46 ± 16 hunting attempts per night, n = 79; Females: 25 ± 11 hunting attempts per nights, n=84; Fig. 3A, Fig S8). However, females were more likely to use a sit and wait strategy than males (females: 24% ± 15%, males: 13% ± 10%, Fig.S9). As a result, the number of perching events per night was similar between males and females (Females: 76 ± 23 perching events per nights; Males: 69 ± 20 perching events per night; Fig S8).” (lines 165 – 174)

Comment 7. In addition, state if the information describes breeding pairs of males and females and provides statistics on the number of tracked pairs and the number of nest boxes.

The revised manuscript now includes a description of the number of tracked breeding pairs and the number of nest boxes. “Of these individuals, 142 belonged to pairs for which data were recovered from both partners (71 pairs in total, 40 in 2019, 31 in 2020). The remaining 21 individuals belonged to pairs with data from one partner (11 females and 1 male in 2019; 4 females and 5 males in 2020).” (lines 82 – 85.)

Comment 8. Line 93: Briefly define the term "landing force" and explain how it was measured (and let the reader know that there is a detailed description in the Methods).

We now include a brief definition of the “landing force” along with a brief explanation of how it was measured in the results section. “We extracted the peak vectoral sum of the raw acceleration during each landing and converted this to ground reaction force (hereafter “landing force”, in Newtons) using measurements of individual body mass (see methods for detailed description).” (lines 92 – 95).

Comment 9. Line 94: All definitions, including "pre-hunting force," need to be better described in the Results section.

Thank you for this suggestion. We now provided a better description of those key definitions directly in the results section:

Measurement of landing force: “Barn owls employing a sit-and-wait strategy land on multiple perches before initiating an attack, with successive landings reducing the distance to the target prey (Fig. 2C).

We used the acceleration data to identify 84,855 landings. These were further categorized into perching events (n = 56,874) and hunting strikes (n = 27,981), depending whether barn owls were landing on a perch or attempting to strike prey on the ground (Fig. 1A and B, see methods for specific details on behavioral classification).” (lines 88 – 95)

Pre-hunt perching force predicts hunting success: “Finally, we analyzed whether the landing force in the last perching event before each hunting attempt (i.e. pre-hunt perching force) predicted variation in hunting success” (lines 229 – 230)

Comment 10. Line 102: Remove "Our analysis of 27,981 hunting strikes showed that" and add "n = 27,981" after the statistics. You have already stated your sample size earlier. There is no need to emphasize it again, although your sample size is impressive.

We modified the text in the results section as suggested.

Comment 11. Line 104: The results so far suggest that the difference in landing force between males and females is an outcome of their different body masses. However, it is not clear what is the reason for the difference in the number of hunting strike attempts between males and females (Lines 104-106). Can you compare the difference in landing force between males and females with similar body mass (females from the lower part of the distribution and males from the upper part)? Is there still a difference?

Thank you, following your comment we made some new analyses that clarified the situation around landing force involved in perching and hunting strike events between sexes. But firstly, we wanted to clarify why there is a difference in number of hunting attempts between males and females. During the breeding season, females typically perform most of the incubation, brooding, and feeding of nestlings in the nest, while the male primarily hunts food for the female and chicks. The female supports the male providing food in a very irregular way, and this changes from pair to pair (paper in prep.). The differences in number of hunting attempts between males and females reflects this asymmetry in food provisioning between sexes during this specific period. We specified this in the revised version of the manuscript (lines 164 – 174).

We also provide a new analysis to investigate sex differences in mass-specific landing force (force/body mass). We found that males and females produce similar force per unit of body mass during perching events. This demonstrates that the overall higher perching force in females (see Fig. 4C in the manuscript) is therefore driven by their higher body mass. (lines 194 – 199)

Comment 12. Line 154: I believe Boonman et al. (2018) is relevant to this part of the discussion. Boonman, Arjan, et al. found that barn owl noise during landing and taking off is worth considering. "The sounds of silence: barn owl noise in landing and taking off."Behavioral Processes 157 (2018): 484-488.

We now cited this paper in the discussion.

Comment 13. Line 164: Your results do not directly demonstrate a link to fitness, although they potentially serve as a proxy for fitness (add a reference). However, you might have information regarding nestlings' survival - that will provide a direct link for fitness. Change your statement or add the relevant data.

We appreciated your feedback, and we adjusted the language accordingly.

Comment 14. Line 213: If the poles are closer to the ground - is it possible that the higher trees and buildings serve for resting and gathering environmental information over greater distances? For example, identifying prey at farther distances or navigating to the next pole?

Yes, this is indeed the most likely explanation for the fact that owls land more on buildings and trees than on poles until the last period (about 6 minutes) before hunting. In these last minutes, barn owls preferentially use poles, as we showed in figure 2B. The revised manuscript now includes this explanation in the discussion (lines 269 – 284).

Comment 15. Line 250: The product "AXY-Trek loggers" does not appear on the Technosmart website (there are similar names, but not an exact match). Are you sure this is the correct name of the tracking device you used?

Thank you for pointing out this detail that we missed. The device we used is now called "AXY-Trek Mini" (https://www.technosmart.eu/axy-trek-mini/). We have corrected this error directly in the revised manuscript.

Comment 16. Line 256: Please explain how the devices were recovered. Did you recapture the animals? If so, how? Additionally, replace "after approximately 15 days" with the exact average and standard deviation. Furthermore, since you have these data, please state the difference in body mass between the two measurements before and after tagging.

The birds were recaptured to recover the devices. Adults barn owls were recaptured at their nest sites, again using automatic sliding traps that are activated when birds enter the nest box. The statement "after approximately 15 days" was replaced by the exact mean and standard deviation, which were 10.47 ± 2.27 days. Those numbers exclude five individuals from the total of 163 individuals included in this study. They could not be recaptured in the appropriate time window but were re-encountered when they initiated a second clutch later in the season (4 individuals) or a new clutch the year after (1 individual).

We integrated this previously missing information in the revised manuscript (lines 370 – 372).

Comment 17. Line 259: What was the resolution of the camera? What were the recording methods and schedule? How did you analyze these data?

The resolution was set to 3.1 megapixel. Motion sensitive camera traps were installed at the entrance to each nest box throughout the period when the barn owls were wearing data loggers, and each movement detected triggered the capture of three photos in bursts. The photos recorded were not analyzed as such for this study, but were used to confirm each supply of prey, which had previously been detected from the accelerometer data. We added these details in the revised manuscript (lines 377 – 380)

Comment 18_1. Figure 1:Panel (A) Include the sex of the described individual.

The sex of the described individual is now included in the figure caption.

Comment 18_2. It would be interesting to show these data for both males and females from the same nest box (choose another example if you don't have the data for this specific nest box).

Although we agree that showing tracks of males and females from the same nest is very interesting, the purpose of this figure was to illustrate our data annotation process and we believe that adding too many details on this figure will make it appear messy. However, the revised manuscript now includes a new figure (Fig. 3A) which shows simultaneous GPS tracks of a male and a female during a complete night, with detailed information about perching and hunting behaviors.

Comment 18_3. Add the symbol of the nest box to the legend.

Done

Comment 18_4. Provide information about the total time of the foraging trip in the text below.

The duration of the illustrated foraging trip has been included in the figure caption.

Comment 18_5. To enhance the figure’s information on foraging behavior, consider color coding the trajectory based on time and adding a background representing the landscape. Since this paper may be of interest to researchers unfamiliar with barn owl foraging behavior, it could answer some common questions.

For similar reasons explained in our answer above (Comment 18_2), we would rather keep this figure as clean as possible. However, we followed your recommendations and included these details in the new Figure 3 described above. In this new figure, GPS tracks are color coded according to the foraging trip number and includes a background representing the landscape. To provide even more detail about the landscape, we added another figure in the supplementary materials (Fig. S2) which provides illustration of barn owls foraging ground and nest site that we think might be of interest for people unfamiliar with barn owls.

Comment 18_6. Inset panels provide a detailed description of the acceleration insert panels.

Done

Comment 18_7. Color code the acceleration data with different colors for each axis, add x and y axes with labels, and ensure the time frame on the x-axis is clear. How was the self-feeding behavior verified (should be described in the methods section)?

We kept both inset panels as simple as possible since they serve here as examples, but a complete representation of these behaviors (with time frame, different colors and labels) is provided in the supplementary materials (figure S3). We included this statement in the figure caption and added a reference to the full representations from the supplementary materials:

In the Figure caption: “Inset panels show an example of the pattern of the tri-axial acceleration corresponding to both nest-box return and self-feeding behaviors (but see Fig S3for a detailed representation of the acceleration pattern corresponding to each behavior).”

In the Method section: “Self-feeding was evident from multiple and regular acceleration peaks in the surge and heave axes (resulting in peaks in VeDBA values > 0.2 g and < 0.9 g, Fig.S3D), with each peak corresponding to the movement of the head as the prey was swallowed whole.”.

Comment 18_8. Panel B Note in the caption that you refer to the acceleration z-axis.

We believe that keeping the statement “the heave acceleration…” in the figure caption is more informative than referring to the “z-axis” as it describes the real dimension to which we are referring. The use of the x, y and z axes can be misleading as they can be interchanged depending on the type and setting of recorders used.

Comment 18_9. Present the same time scale for both hunting strategies to facilitate comparison. You can achieve this by showing only part of the flight phase before perching.

Done

Comment 18_10. Panel C Presenting the data for both hunting strategy and sex would provide more comprehensive information about the results and would be relatively easy to implement.

We agree with your comment. We present the differences in landing force for both landing contexts and sexes in the new Figure 3 as well as in the supplementary materials (Figure S10) of this revised manuscript.

Comment 19. Figure 2: Please provide an explanation of the meaning of the circles in the figure caption.

Done

Comment 20. Figure 3:Panel A It is unclear how the owl illustration is relevant to this specific figure, unlike the previous figures where it is clear. Also, suggest removing the upper black line from the edge of the figure or add a line on the right side.

Done (now in Figure 2).

Panel (B) "Density" should be capitalized.

Done

Panel (C) Add a scale in meters, and it would be helpful to include an indication of time before hunting for each data point.

Done

Comment 21. Figure S1: Mark the locations of the nest boxes and ensure that trajectories of different individuals and sexes can be identified.

The purpose of this figure was to show the spatial distribution of the data. We think that adding nest locations and coloring the paths according to individuals and/or sex will make the figure less clear. However, the new Figure 3 highlights those details.

Comment 22. Figure S2: Show the pitch angle similarly to how you showed the acceleration axes, and explain what "VeDBA" stands for. Provide a description of the perching behavior, clearly indicating it on the figure. Add axes (x, y, z) to the illustration of the acceleration explanation.

We edited this figure (now figure S3) to show the pitch angle and provide an explanation of what “VeDBA” stands for in the figure caption. The figure caption now also provides a better description of the perching behavior. For the axes (i.e. X, Y, Z), we prefer to refer to the heave, surge, and sway as this is more informative and refers to what is usually reported in studies working with tri-axial accelerometers.

Comment 23. Table S1: Improve the explanation in the caption and titles of the table.

Done

**Reviewer #2 (Recommendations For The Authors):**
Comment 1. From the public review and my assessment there, the authors can be assured that I thoroughly enjoyed the read and am looking forward to seeing a revised and improved version of this paper.

We thank the reviewer for this comment. We revised the manuscript according to their comments.

Comment 2. In addition to my major points stated above, I would like to add the following recommendations:The manuscript is overall well written, but it uses a very pictorial language (a little as if we were in a David Attenborough documentary) that I find inappropriate for a research paper especially in the abstract and introduction, "remarkable" (2x), "sophisticated" (are there any unsophisticated adaptations? We are referring to something under selection after all) etc.

We appreciated that you found the paper overall well written, and we understand the comment about pictorial language. We therefore slightly changed the text to make sure that the adjective used to describe adaptive strategies are not over-emphasized.

Comment 3. Abstract"While the theoretical benefits of predator camouflage are well established, no study has yet been able to quantify its consequences for hunting success." - This claim is actually not fully true:Nebel Carina, Sumasgutner Petra, Pajot Adrien and Amar Arjun 2019: Response time of an avian prey to a simulated hawk attack is slower in darker conditions, but is independent of hawk colour morph. Soc. open sci.6:190677

We edited our claim to specify that the consequences of predator camouflage on hunting success has never been quantified in natural conditions and cited the reference in the introduction.

Comment 4. Line 23. Rephrase to: "We used high-resolution movement data to quantify how barn owls (Tyto alba) conceal their approach when using a sit-and-wait strategy, as well as the power exerted during strikes."

We edited this sentence in the abstract, as suggested.

Comment 5. ResultsThere is a disconnect between the objectives outlined at the end of the introduction and the following results that should be improved.The authors state: "Using high-frequency GPS and accelerometer data from wild barn owls (Tyto alba), we quantify the landing dynamics of this sit-and-wait strategy to (i) examine how birds adjust their landing force with the behavioral and environmental context and (ii) test the extent to which the magnitude of the predator cue affects hunting success." But one of the first results presented are sex differences.

This is a fair point. We have now changed our statement in the end of the introduction as well as the order of the results to improve the link between the objectives outlined in the introduction and the way result are presented.

Comment 6. At this stage, the reader does not even know yet that we are presented with a size-dimorphic species that also has very different parental roles during the breeding season. This should be better streamlined, with an extra paragraph in the introduction. And these sex differences are then not even discussed, so why bring them up in the first place (and not just state "sex has been fitted as additional co-variate to account for the size-dimorphism in the species" without further details).

We edited the way the objectives are outlined in the introduction to cover the size dimorphism (lines 70 – 76). We also completely changed the way the sex differences are presented in the results, including a new analysis that we believe provides a better comprehensive understanding of barn owl foraging behavior (lines 164 – 206). Finally, we added a new paragraph in the discussion to consider those results (lines 319 – 339).

Comment 7. It is not clear to me where and how high-resolution GPS data were used? The results seem to concentrate on ACC – why GPS was used and how it features should be foreshadowed in a few lines in the introduction. I definitively prefer having the methods at the end of a manuscript, but with this structure, it is crucial to give the reader some help to understand the storyline.

GPS data were used to validate some behavioral classifications (prey provisioning for example), but most importantly they were used to link each landing event with perch types. We edited the text in the result section to clarify where GPS and/or ACC data were used.

Comment 8. DiscussionMove the orca example further down, where more detail can be provided to understand the evidence.

After our extensive edits in the discussion, we felt this example was interrupting the flow. We now cite this study in the introduction.

Comment 9. Size dimorphism and evident sex differences are not discussed.

The revised manuscript now includes a new paragraph in the discussion in which sex differences are discussed (lines 319 – 339).

Comment 10. Be more precise in the terminology used (for example, land use seems to be interchangeable with habitat characteristics?).

We modified “land use” with “habitat data” in the revised manuscript.

Comment 11. MethodsPlease provide a justification for the very high weight limit (5%; line 256). This limit is outdated and does not fulfill the international standard of 3% body weight. I assume the ethics clearance went through because of the short nature of the study (i.e., the birds were not burdened for life with the excess weight? But a line is needed here or under the ethics considerations to clarify this).

The 5% weight limit was considered acceptable due to the short deployment period, and we now edited the ethics statement to emphasize this point. However, it is important to note that there is no real international standard, with both 3% and 5% weight limits being commonly used. Both limits are arbitrary and the impact of a fixed mass on a bird varies with species and flight style. All owls survived and bred similarly to the non-tagged individuals in the population (lines 373 – 376 & lines 558 – 561)

EDITORIAL COMMENT: We strongly encourage you to provide further context and clarification on this issue, as suggested by the Reviewer. On a related point, the ethics statement refers to GPS loggers, rather than GPS and ACC devices; we encourage you to clarify wording here.

Thank you for highlighting this point that indeed needed some clarifications.

Although we have used the terminology "GPS recorders", the authorization granted by the Swiss authorities for this study effectively covers the entire tracking system, which combines both GPS and ACC recorders in the same device. We have therefore changed the wording used in the ethics statement to avoid any misunderstanding (lines 373 – 376 & lines 558 – 561)

Comment 12. Please provide more information on the model selection approach, what does "Non-significant terms were dropped via model simplification by comparing model AIC with and without terms." mean? Did the authors use a stepwise backward elimination procedure (drop1 function)? Or did they apply a complete comparison of several candidate models? I think a model comparison approach rather than stepwise selection would be more informative, as several rather than only one model could be equally probable. This might also improve model weights or might require a model averaging procedure - current reported R2values are very small and do not seem to support the results well.

We apologize for the lack of details about this important aspect of the statistical analysis. We applied an automated stepwise selection using the *dredge* function from the R package “MuMin”, therefore applying a complete comparison of several candidate models. The final models were chosen as the best models since the number of candidate models within ∆AIC<2 was relatively low in each analysis and thus a model averaging was not appropriate here. We edited the methods section to ensure clarity, and added model selection tables for each analysis, ranked according to AICc scores, in the supplementary materials (lines 532 – 552)

In addition, we agree that the reported R-squared values in our analyses are quite low, specifically regarding the influence of pre-hunt perching force on hunting success (cond R2 = 0.04). Nonetheless, landing impact still has a notable effect size (an increase of 1N reduces hunting success by 15%). The reported values are indicative of the inherent complexity in studying hunting behavior in a wild setting where numerous variables come into play. We specifically investigated the hypothesis that the force involved during pre-hunt landings, and consequently the emitted noise, influences the success of the next hunting attempt in wild barn owls. Factors such as prey behavior and micro-habitat characteristics surrounding prey (such as substrate type and vegetation height) are most likely to be influential but hard, or nearly impossible, to model. We now cover this in a more nuanced way in the discussion (lines 266 – 268)

Comment 13. Please explain why BirdID was nested in NightID - this is not clear to me.

Probably here there is a misunderstanding because we wrote that we nested NightID in BirdID (and not BirdID in NightID).

Comment 14. I hope the final graphs and legends will be larger, they are almost impossible to read.

We enlarged the graphs and legends as much as possible to improve readability. However, looking at the graphs in the published version they seem clear and readable.

Comment 15. Figure S1: Does "representation" mean the tracks don't show all of the 163 owls? If so, be precise and tell us how many are illustrated in the figure.

Figure S1 represent the tracks for each of the 163 barn owls used in the study. We changed the terminology used in the figure caption to avoid any misunderstanding.

Comment 16. Figure S4: Please adjust the y-axis to a readable format.

Done